# Structures of three MORN repeat proteins and a re-evaluation of the proposed lipid-binding properties of MORN repeats

Sara Sajko[1][☮], Irina Grishkovskaya[1][☮], Julius Kostan[1][☮], Melissa Graewert[2], Kim Setiawan[3], Linda Trübestein[1], Korbinian Niedermüller[3], Charlotte Gehin[4,5], Antonio Sponga[1], Martin Puchinger[1], Anne-Claude Gavin[4,6], Thomas A. Leonard[1], Dimitri I. Svergun[2], Terry K. Smith[7], Brooke Morriswood[3]*, Kristina Djinovic-Carugo[1,8]*

1 Department of Structural and Computational Biology, Max Perutz Labs, University of Vienna, Vienna, Austria, 2 European Molecular Biology Laboratory, Hamburg Unit, Hamburg, Germany, 3 Department of Cell and Developmental Biology, Biocenter, University of Würzburg, Würzburg, Germany, 4 European Molecular Biology Laboratory, Heidelberg Unit, Heidelberg, Germany, 5 Institute of Bioengineering, Laboratory of Lipid Cell Biology, École Polytechnique Fédérale de Lausanne (EPFL), Lausanne, Switzerland, 6 Department for Cell Physiology and Metabolism, University of Geneva, Centre Medical Universitaire, Geneva, Switzerland, 7 School of Biology, BSRC, University of St. Andrews, St. Andrews, United Kingdom, 8 Department of Biochemistry, Faculty of Chemistry and Chemical Technology, University of Ljubljana, Ljubljana, Slovenia

☮ These authors contributed equally to this work.
* brooke.morriswood@uni-wuerzburg.de (BM); kristina.djinovic@univie.ac.at (KDC)

**Data Availability Statement:** Structure files have been uploaded to PDB (accession codes 6T4D, 6T4R, 6T68, 6T69, 6T6Q) and SASBDB (accession codes SASDG97, SASDGA7, SASDGB7,

## Abstract

MORN (Membrane Occupation and Recognition Nexus) repeat proteins have a wide taxonomic distribution, being found in both prokaryotes and eukaryotes. Despite this ubiquity, they remain poorly characterised at both a structural and a functional level compared to other common repeats. In functional terms, they are often assumed to be lipid-binding modules that mediate membrane targeting. We addressed this putative activity by focusing on a protein composed solely of MORN repeats—*Trypanosoma brucei* MORN1. Surprisingly, no evidence for binding to membranes or lipid vesicles by TbMORN1 could be obtained either in vivo or in vitro. Conversely, TbMORN1 did interact with individual phospholipids. High-and low-resolution structures of the MORN1 protein from *Trypanosoma brucei* and homologous proteins from the parasites *Toxoplasma gondii* and *Plasmodium falciparum* were obtained using a combination of macromolecular crystallography, small-angle X-ray scattering, and electron microscopy. This enabled a first structure-based definition of the MORN repeat itself. Furthermore, all three structures dimerised via their C-termini in an antiparallel configuration. The dimers could form extended or V-shaped quaternary structures depending on the presence of specific interface residues. This work provides a new perspective on MORN repeats, showing that they are protein-protein interaction modules capable of mediating both dimerisation and oligomerisation.

SASDGC7) All other relevant data are within the paper and its Supporting Information files.

**Funding:** Austrian Science Fund (FWF):Julius Kostan,Brooke Morriswood,Kristina Djinovic-Carugo P22265-B12; Austrian Science Fund (FWF):Sara Sajko,Irina Grishkovskaya,Julius Kostan,Kim Setiawan,Korbinian Niedermuller,Brooke Morriswood,Kristina Djinovic-Carugo P27016-B21; Austrian Science Fund (FWF):Linda Truebestein, Thomas A Leonard P28135; Austrian Science Fund (FWF):Linda Truebestein,Thomas A Leonard P30584; Austrian Science Fund (FWF):Sara Sajko, Irina Grishkovskaya,Julius Kostan,Antonio Sponga, Martin Puchinger,Kristina Djinovic-Carugo I525; Austrian Science Fund (FWF):Sara Sajko,Irina Grishkovskaya,Julius Kostan,Antonio Sponga, Martin Puchinger,Kristina Djinovic-Carugo I1593; Austrian Science Fund (FWF):Sara Sajko,Irina Grishkovskaya,Julius Kostan,Antonio Sponga, Martin Puchinger,Kristina Djinovic-Carugo P22276; Austrian Science Fund (FWF):Sara Sajko,Irina Grishkovskaya,Julius Kostan,Antonio Sponga, Martin Puchinger,Kristina Djinovic-Carugo P19060; Austrian Science Fund (FWF):Kristina Djinovic-Carugo W1221; Austrian Federal Ministry of Economy, Family and Youth (BMWFJ):Antonio Sponga,Martin Puchinger,Kristina Djinovic-Carugo 253275; Wellcome Trust (WT):Antonio Sponga, Martin Puchinger,Kristina Djinovic-Carugo 201543/Z/16; COST action:Antonio Sponga,Martin Puchinger,Kristina Djinovic-Carugo BM1405; Vienna Science and Technology Fund (WWTF): Sara Sajko,Irina Grishkovskaya,Julius Kostan, Antonio Sponga,Martin Puchinger,Kristina Djinovic-Carugo LS17-008; Universität Wien (University of Vienna):Sara Sajko,Irina Grishkovskaya,Julius Kostan,Antonio Sponga, Martin Puchinger,Kristina Djinovic-Carugo; Louis-Jeantet Foundation (Fondation Louis-Jeantet): Charlotte Gehin,Anne-Claude Gavin; UK Research and Innovation | Medical Research Council (MRC): Terry K. Smith MR/Mo20118/1.

**Competing interests:** The authors have declared that no competing interests exist.

## Introduction

MORN (Membrane Occupation and Recognition Nexus) repeats were first discovered in 2000, as a result of a screen for proteins present in the triad junctions of skeletal muscle [1]. The junctophilins, the protein family identified in this screen, were observed to have 8 repeats present in their N-terminal regions. The repeats were given the name MORN based on a proposed role in mediating plasma membrane association of the N-terminal domain of the junctophilins. The MORN repeats were initially classified as being 14 amino acids in length, with an approximate consensus sequence of YEGEWxNGKxHGYG [1]. A bioinformatics analysis at the time indicated that assemblies of 8 consecutive MORN repeats were also present in a putative junctophilin orthologue in a nematode (*Caenorhabditis elegans*), a family of plant (*Arabidopsis thaliana*) lipid kinases, and a bacterial (*Cyanobacterium*) protein [1]. Later genome-era bioinformatics has shown that MORN repeat proteins are in fact found ubiquitously, being present in both eukaryotes and prokaryotes [2].

The number of MORN repeats in any given protein can vary greatly, from 2 to over 20, and they are found in combination with a wide range of other domains. Most published work now favours a 23-amino acid length for a single MORN repeat, with the highly-conserved GxG motif at residues 12–14 [3–5]. A 14-amino acid length is still favoured by some groups, however [6, 7]. Notable mammalian MORN repeat proteins besides the junctophilins include ALS2/alsin, at least two radial spoke proteins (RSPH10B, RSPH1/meichroacidin), the histone methyltransferase SETD7, and MORN4/retinophilin [8–11].

MORN repeats are generally assumed to be lipid-binding modules, but direct evidence for this function is actually lacking. In junctophilins, there is good evidence that the N-terminal region containing the MORN repeats mediates plasma membrane targeting [1, 12, 13]. It has not been demonstrated whether the MORN repeats or the other sequences in the N-terminal region are responsible for this targeting activity. In addition, the targeting activity itself may be due to either protein-lipid or protein-protein interactions.

Although there is good evidence that the N-terminal region (amino acids 1–452), of junctophilin-2 can directly bind lipids, it has not specifically been shown that the MORN repeats are responsible [7]. This is because the N-terminal region consists of 6 MORN repeats, a 140 amino acids long joining region, 2 additional MORN repeats, and an alpha-helical region. Binding could therefore be mediated by other nearby sequences, especially the run of over 100 amino acids that occurs between repeats 6 and 7. Work on the family of plant phosphatidylinositol(4)phosphate 5-kinases (PIPKs) that contain MORN repeats has led to suggestions that the repeats might regulate the activity of the kinase domain, bind to phospholipids, or mediate protein-protein interactions [5, 14, 15]. Another recent report on junctophilin-2 showed that upon cleavage by an endogenous protease, the N-terminal region translocates to the nucleus via a nuclear localisation signal and functions there as a transcription factor [16]. It therefore remains unclear what role(s) this ubiquitous class of repeat actually have [17].

Coupled with this lack of unambiguous functional data is a lack of high-resolution structural information, exemplified by the ongoing lack of consensus as to whether a single repeat is composed of 14 or 23 amino acids. This contrasts sharply with the considerable amount of information available on other classes of protein repeats such as ankyrin repeats, leucine-rich repeats, or WD40 repeats [18, 19]. Until very recently, the structure of the histone methyltransferase SETD7 was the sole representative of the MORN repeat protein family [9, 20, 21]. Even here, the structure of the N-terminal domain containing the MORN repeats is incomplete, and the level of sequence similarity of the repeats to those of junctophilins and other MORN repeat-containing proteins makes assignment difficult. Each repeat appears to form a β-hairpin with an acidic surface, but it remains unclear if this is a general property of MORN repeats.

The SETD7 structure has not been analysed in this context, with more focus applied to its catalytic methyltransferase domain. In particular, a structure-based definition of the repeat class itself is currently lacking.

To address this, and additionally to tackle the question of putative lipid binding, it would obviously be advantageous to utilise a protein that is composed solely of MORN repeats. In this way, the contribution of other sequences or domains could be discounted. The MORN1 protein from the early-branching eukaryote *Trypanosoma brucei* is an ideal candidate in this regard, and has the advantage of also being well-characterised at a cell biology level [22–25].

TbMORN1 consists of 15 consecutive 23-amino acid MORN repeats, with barely any intervening sequence (Fig 1A). In *T. brucei*, TbMORN1 is localised to a ~2 μm long cytoskeleton-associated structure (the hook complex) that is found just below the inner leaflet of the plasma membrane. The hook complex encircles the neck of a small invagination of the plasma membrane that contains the root of the cell's single flagellum [26]. This invagination, termed the flagellar pocket, is the sole site of endo- and exocytosis in trypanosomes and is thought to be analogous to the ciliary pocket that is found at the base of some mammalian primary cilia [27–29].

Previous work on TbMORN1 demonstrated by fluorescence recovery after photobleaching (FRAP) that it is a stable component of the hook complex and a list of its binding partners and near neighbours has been obtained using proximity-dependent biotin identification (BioID) [23, 24]. In functional terms, depletion of TbMORN1 by RNAi in the mammalian-infective

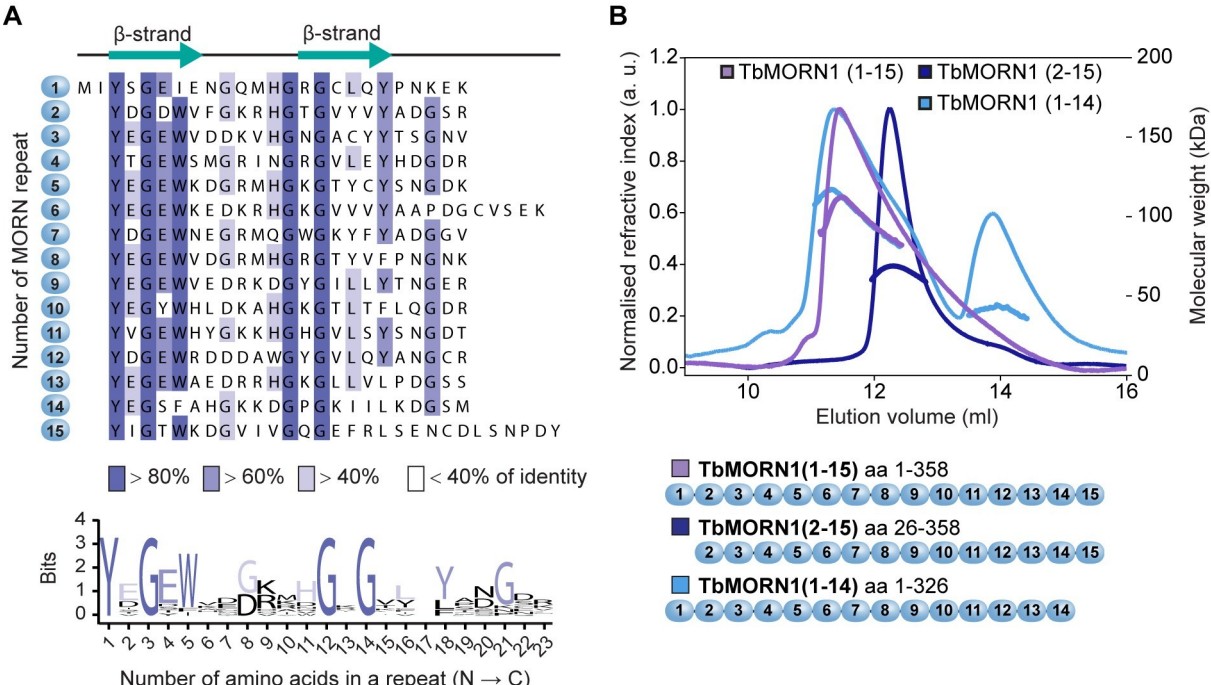

**Fig 1. TbMORN1 primary structure and dimerisation. (A)** Primary structure of TbMORN1, with individual MORN repeats shown in alignment, and coloured according to amino acid conservation. A schematic of the predicted secondary structure of each repeat is shown above the alignment. A consensus amino acid sequence of the individual MORN repeats from TbMORN1 based on the alignment is shown in the sequence logo below. **(B)** TbMORN1 dimerises via its C-terminus. SEC-MALS profiles of TbMORN1(1–15), TbMORN1(2–15), and TbMORN1(1–14). Schematics are shown underneath. TbMORN1(1–15) tended to form high-order assemblies, whereas removal of the first MORN repeat resulted in a monodisperse dimer. Removal of the last MORN repeat in TbMORN1(1–14) resulted in a polydisperse mixture of monomers, dimers, and other species. Chromatographic separation was done using a Superdex 200 Increase 10/300 GL column, void volume 7.2 ml.

(bloodstream) form of the parasite resulted in a lethal phenotype [22]. Functional analysis indicated that the protein might be involved in endocytosis, as well as regulating the flow of macromolecular cargo through the neck of the flagellar pocket [25].

In order to investigate the putative lipid-binding capacity of TbMORN1 and its three-dimensional structure, a detailed biochemical, structural, and functional analysis of the TbMORN1 protein was carried out. A truncated form missing the first MORN repeat that was best suited for in vitro work was found not to bind to phospholipid vesicles under any conditions but was able to associate with individual lipid molecules. High-resolution crystal structures and biophysical analyses of a truncated form of the TbMORN1 protein and its homologues from the parasites *Toxoplasma gondii* (TgMORN1) and *Plasmodium falciparum* (PfMORN1) enabled, for the first time, a structure-based definition of the MORN repeat. All three structures are antiparallel dimers joined via their C-termini, and, when possessed of the correct key residues, can adopt extended or V-shaped quaternary structures. This provides a new insight into MORN repeats, showing that they are capable of mediating homotypic interactions.

## Materials and methods

### Antibodies and other reagents

All custom antibodies have been described previously. The rabbit anti-TbMORN1 were made for a previous project [24]. The mouse monoclonal anti-Ty1 (BB2) antibodies were a gift from Cynthia He (University of Singapore) [30]. The mouse monoclonal anti-PFR1,2 antibodies (L13D6) were a gift from Keith Gull (University of Oxford) [31]. The anti-BiP antibodies were a gift from Jay Bangs (University at Buffalo) [32]. The rabbit anti-GFP antibodies were a gift from Graham Warren (MRC Laboratory for Molecular Cell Biology) [33]. The following antibodies were obtained from commercial sources: anti-strep tag StrepMAB-Classic (IBA Biosciences), HRP-conjugated anti-mouse (Thermo Fisher Scientific), anti-GST (Santa Cruz Biotechnology). Defatted BSA was purchased from Sigma-Aldrich.

### Cloning and mutagenesis of expression constructs

The 1077 bp TbMORN1 open reading frame (ORF) (UniProt accession No. Q587D3; Tri-TrypDB database accession No. Tb927.6.4670) was amplified by PCR from genomic DNA obtained from *Trypanosoma brucei brucei* strain Lister 427 and ligated into vector pETM-13 encoding a Strep-tag at the 3' end of the insert. TbMORN1 truncations were generated using this construct as a template by ligase-independent cloning [34]. The sequences for TbMORN1 (2–15) and TbMORN1(1–14) were additionally ligated into the pCoofy12 vector encoding a 3C protease-cleavable N-terminal Twin-Strep-tag [35] by sequence and ligation-independent cloning [36]. Mutagenesis constructs were generated by standard methods using the pre-existing pCoofy12_TbMORN1(2–15) construct as the template [37]. All primer sequences are available upon request. For LiMA experiments, the construct encoding the EGFP-TbMORN1 (2–15) was cloned in a two-step procedure by sequence and ligation-independent cloning followed by Gibson assembly using pCoofy12_TbMORN1(2–15) and the pEGFP-C1 vector [38]. The 1092 bp TgMORN1 (UniProt accession No. Q3S2E8) and 1095 bp PfMORN1 (UniProt accession No. Q8IJ93) ORFs were amplified by PCR from genomic DNA. Truncations of the TgMORN1 and PfMORN1 constructs were generated using ligase-independent cloning. The TgMORN1 constructs were additionally ligated into the pET14 vector encoding a 3C protease-cleavable N-terminal His10-tag. The PfMORN1 constructs were additionally ligated into the pCoofy32 encoding a 3C protease-cleavable N-terminal His10-tag and C-terminal OneStrep-tag.

## Recombinant protein expression and purification

Rosetta 2 (DE3)pLysS bacterial cells transformed with the required expression plasmids were grown at 37˚C with shaking in the presence of the appropriate antibiotics. Large scale expression was carried out either in Luria-Broth or in auto-induction (ZY) medium [39], with 500 ml media being inoculated with 3–5 ml of pre-cultured cells. Cells in Luria-Broth were grown to an $OD_{600}$ ~ 0.8–1.0, after which 50 μM IPTG was added to induce recombinant protein expression. The cells were then incubated at lower temperature (overnight, 20˚C). The cells were then harvested by centrifugation (5000 x g, 30 min), and either lysed immediately or stored at -80˚C. For purification, the cells were resuspended in lysis buffer (50 mM Tris-HCl pH 8.5, 200 mM NaCl, 5% (w/v) glycerol, 1 mM DTT, protease inhibitor cocktail, benzonase). The pellet emulsions were first homogenised by mixing on ice using a T 10 basic Ultra-Turrax dispersing instrument (IKA), and lysis was accomplished using a single cycle in a cell disruptor (Constant systems Ltd), with the pressure set to 1.35 kPa. Lysates were clarified by centrifugation (18,000 x g, 45 min, 4˚C), and a two-step fast protein liquid chromatography (FPLC) purification protocol using an ÄKTA Protein Purification System (GE Healthcare Life Sciences) at 8˚C was then followed to obtain the recombinant protein. The supernatants were applied to two connected Strep-Trap HP 5 ml columns packed with Strep-Tactin ligand immobilized in an agarose matrix (GE Healthcare Life Sciences) previously equilibrated with equilibration buffer (50 mM Tris-HCl pH 8.5, 200 mM NaCl, 2% (w/v) glycerol, 1 mM DTT). Flow speed was adjusted to 2.5 ml/min. When 100% step gradient of elution buffer (equilibration buffer plus 2.5 mM D-desthiobiotin) was applied, the bound proteins were eluted in a single chromatographic peak. Selected peak fractions were examined by SDS-PAGE for protein content and purity, pooled accordingly, and concentrated in Amicon Ultra centrifugal filter units (MerckMillipore, various pore sizes) according to the manufacturer's instructions. These affinity-purified protein concentrates were then applied to a previously equilibrated HiLoad 16/600 Superdex 200 pg column (GE Healthcare Life Sciences) packed with dextran covalently bound to highly cross-linked agarose, enabling separation of proteins with MW in the range of 10–600 kDa. Flow speed was adjusted to 1 ml/min and fractions of 1.5 ml were collected. Fractions corresponding to the targeted chromatographic peak were examined for protein content by SDS-PAGE, pooled accordingly to their purity, concentrated, and stored at -80˚C until use.

## Limited proteolysis

Purified recombinant His-TbMORN1(1–15) at 1 mg/ml was separately incubated with three proteases with different cleavage specificities (α-chymotrypsin, trypsin, and proteinase K) in 20 mM Tris-HCl pH 8.5, 200 mM NaCl, 2% glycerol, 0.2 mM $CaCl_2$ (15 min, room temperature (RT)). The proteases were used at dilutions of 1:100–1:2000. The reactions were stopped by the addition of SDS-Coomassie sample loading buffer for analysis by gel electrophoresis. The indicated protein bands were extracted from the gel and subjected to mass spectrometry analysis.

## Size-exclusion chromatography coupled to multi-angle light scattering (SEC-MALS)

The MW and oligomeric state of purified proteins were verified by size exclusion chromatography (SEC) coupled to multi-angle light scattering (MALS), using a Superdex 200 Increase 10/300 GL column (GE Healthcare Life Sciences). Up to five protein samples of 100 μl were dialysed against 2x 1 L of freshly-prepared, degassed gel filtration buffer (20 mM Tris-HCl pH 8.5, 200 mM NaCl, 2% (w/v) glycerol, 0.5 mM DTT) (overnight, 4˚C). Gel filtration buffer was

also used for overnight equilibration of the column and in the subsequent measurements. Protein samples were clarified by centrifugation using a TLA-55 rotor in an Optima MAX-XP table top ultracentrifuge (Beckman Coulter) (90,720 x g, 30 min, 4˚C). 100 μl of 2–4 mg/ml protein samples were applied to a column using the 1260 Infinity HPLC system (Agilent Technologies) coupled to a MiniDawn Treos detector (Wyatt Technologies) with a laser emitting at 690 nm. An RI-101 detector (Shodex) was used for refractive index determination and the Astra 7 software package (Wyatt Technologies) for data analysis. No correction of the refractive index was necessary due to the 2% (w/v) glycerol content in the buffer.

## Circular Dichroism (CD)

Far-UV CD was used both for measurement of secondary structure and for validation of the thermostability of TbMORN1 constructs. To avoid the absorption of Tris and NaCl below 180 nm [40], three protein samples of 100 μl were first dialysed against 2x 1 L of dialysis buffer (20 mM $NaH_2PO_4$, 20 mM $Na_2HPO_4$, 200 mM NaF) (overnight, 4˚C). The pH 8.0 was adjusted by mixing the mono- and dibasic sodium phosphate solutions. The dialysed proteins were clarified by centrifugation in an Optima MAX-XP tabletop ultracentrifuge (Beckman Coulter Life Sciences) (90,720 x g, 30 min, 4˚C). The concentration of protein samples was adjusted to 0.25 mg/ml. CD measurements were carried out in a quartz cuvette with an optical path length of 0.5 mm (Stana Scientific Ltd) using a Chirascan Plus spectrophotometer (Applied Photophysics) equipped with the Chirascan-plus DMS software package. The CD profiles for secondary structure calculations were obtained at RT in the range of 190–260 nm. Further analysis was carried out using the BeStSel server, which is specialised in the analysis of CD data from proteins rich in β-strands [41, 42]. Data were converted to $\Delta\epsilon$ ($M^{-1}cm^{-1}$) and uploaded to the BeStSel online server. Melting experiments were performed in the range of 190–260 nm, 20–80˚C, with a temperature ramp of 0.8˚C/min. Data were analysed with Global 3 software package.

## Chemical cross-linking coupled to mass spectrometry (XL-MS)

For chemical cross-linking with EDC (1-ethyl-3-(3-dimethylaminopropyl)carbodiimide hydrochloride) or $BS^3$ (bis(sulfosuccinimidyl)suberate), 200–300 μl of approximately 30 μM TbMORN1 sample was dialysed twice against 1 L of EDC buffer (20 mM MES-NaOH pH 6.8, 200 mM NaCl) or $BS^3$ buffer (20 mM HEPES-NaOH pH 8.0, 200 mM NaCl) (overnight, 4˚C). Following dialysis, the protein was clarified by centrifugation (90,720 x g, 30 min, 4˚C) using a TLA-55 rotor in an Optima MAX-XP tabletop ultracentrifuge (Beckman Coulter Life Sciences). EDC (Thermo Fisher Scientific) was first equilibrated to RT and then a stock solution in EDC buffer was prepared. 1.3 μM TbMORN1 previously dialysed in EDC buffer was mixed with 0, 200, and 400 μM EDC (all final concentrations) in a total volume of 40 μl. After a 30 min incubation at RT, 10 μl of SDS loading buffer was added and the mixtures were further denatured by heating (95˚C, 10 min). The experiments with $BS^3$ were carried out identically except that $BS^3$ buffer and 3.4 μM TbMORN1 were used, and the incubation time was 120 min. Samples were separated by SDS-PAGE, stained with Coomassie dye, and selected bands corresponding to monomers and dimers cross-linked with 400 μM cross-linker were excised and subjected to enzymatic digestion and subsequent mass spectrometry analysis. Coomassie Brilliant Blue-stained excised bands were destained with a mixture of acetonitrile and 50 mM ammonium bicarbonate (ambic), in two consecutive steps (each 10 min, RT). The proteins were reduced using 10 mM DTT in 50 mM ambic for (30 min, 56˚C), alkylated with 50 mM iodoacetamide in 30 mM ambic in the dark (30 min, RT), and digested with trypsin (Promega, mass spectroscopy grade) (overnight, 37˚C). The reaction was stopped using 10% (v/v) formic acid and extracted peptides were desalted using C18 Stagetips [43]. Peptides were analysed on

an UltiMate 3000 HPLC RSLCnano system coupled to a Q Exactive HF mass spectrometer, equipped with a Nanospray Flex ion source (all Thermo Fisher Scientific). Peptides were loaded onto a trap PepMap 300 C18 column of dimensions 5 mm x 300 μm i.d., packed with 5 μm particles with a pore size of 100 Å (Thermo Fisher Scientific) and separated on an analytical C18 100 column of dimensions 500 mm x 75 μm i.d., packed with 2 μm particles with a pore size of 100 Å (Thermo Fisher Scientific), applying a linear gradient from 2% to 40% solvent B (80% acetonitrile, 0.1% formic acid) at a flow rate of 230 nl/min over 120 min. The mass spectrometer was operated in a data-dependent mode at high resolution of both MS1 and MS2 level. Peptides with a charge of +1, +2 or of a higher than +7, were excluded from fragmentation. To identify cross-linked peptides, the spectra were searched using pLink software v1.23 (Yang et al., 2012). Q Exactive HF raw-files were pre-processed and converted to mgf-files using pParse [44]. The MaxQuant database [45] was used to search the spectra for the most abundant protein hits. Carbamidomethylation of cysteine and oxidation of methionine residues were set as variable modifications. Trypsin was set as an enzyme specificity and EDC or $BS^3$ was set as a cross-linking chemistry. In the case of EDC, aspartic, and glutamic acid residues, as well as C-termini of proteins, were allowed to be linked with lysine residues. In the case of $BS^3$, lysine residues and N-termini of proteins were allowed to be linked with lysine residues, N-termini of proteins, as well as to serine, threonine and tyrosine residues. Search results were filtered for 1% FDR (false discovery rate) on the PSM (number of peptide-spectrum matches) level and a maximum allowed precursor mass deviation of 5 ppm. To remove low quality PSMs, an additional e-Value cutoff of $< 0.001$ was applied. In order to distinguish intra- from inter-molecular chemical cross-links, results from monomers and dimers were compared. A potential inter-molecular cross-link must have shown the following criteria: (1) minimally 3 peptide PSMs in dimer and (2) minimally 3-times more PSMs in dimer than in monomer.

## Protein-lipid overlay assays (PIP strips)

PIP strips were purchased from Echelon Biosciences. PBS-T (PBS, 0.1% TWEEN-20) was used as a general buffer. Purified recombinant TbMORN1(1–15) was clarified by centrifugation (20,817 x g, 20 min, 4˚C) prior to use. The PIP strips were blocked using blocking buffer (3% (w/v) defatted BSA, PBS-T) (60 min, RT) and then incubated with 5 μg/ml of TbMORN1 in 10 ml of blocking solution (60 min, RT). After three washes with PBS-T, the membranes were overlaid with anti-strep antibodies diluted in blocking solution (60 min, RT). After a further three PBS-T washes, the membranes were overlaid with HRP-conjugated secondary antibodies (60 min, RT). The membranes were then washed three times with PBS-T and visualised by ECL (Western Blotting substrate, Thermo Fisher Scientific) using a Fusion FX imager (Vilber Lourmat). All binding and wash steps were carried out with gentle agitation of the membranes. For the positive control, the PIP strip was overlaid with GST-tagged PLC-δ1 PH domain (Echelon Biosciences) and mouse monoclonal anti-GST antibodies (Santa Cruz Biotechnology) were used.

## Fluorescence anisotropy

Stocks of BODIPY TMR-labelled PI C6, PI(4)P C6, PI(3,4)$P_2$ C6, PI(3,5)$P_2$ C6, PI(4,5)$P_2$ C6 and PI(4,5)$P_2$ C16 were sonicated (5 min, RT) in a sonication bath, and in parallel with purified recombinant TbMORN1(2–15), TbMORN1(7–15), TbMORN1(10–15), were clarified by centrifugation (20,817 x g, 20 min, 4˚C). The concentrations of lipid stocks were determined with a Hitachi U-3501 UV-VIS spectrophotometer, using quartz absorbance cuvettes and an optical path length of 10 mm (Hellma Analytics). For this purpose, the maximum absorbance

of BODIPY TMR dye at λ = 544 nm was measured; its extinction coefficient ε = 60,000 cm$^{-1}$M$^{-1}$. The total volume of each respective sample in a cuvette was 110 µl, which consisted of a lipid stock and a protein in gel filtration buffer (20 mM Tris-HCl pH 8.5, 200 mM NaCl, 2% (w/v) glycerol, 0.5 mM DTT). The concentration of selected TbMORN1 constructs was varied from 0 to 35 µM (or more), while the concentration of added lipid was kept constant at 0.1 µM. Measurements were performed on a Perkin Elmer LS50B fluorimeter in quartz cuvettes with an optical path length of 10 x 2 mm (Hellma Analytics). To ensure a constant temperature of 20˚C in the measured sample, the measurement cell was connected to a water bath. The parameters for fluorescence anisotropy (r) measurements were: λ$_{ex}$ = 544 nm and aperture of excitation slit = 15 nm; λ$_{em}$ = 574 nm and aperture of emission wavelength = 20 nm; time of integration = 1 s and T = 20˚C. The grating factor (G factor), which provides grating correction for the optical system, was determined on samples with exclusively 0.1 µM lipid and kept constant during measurement of each concentration series. Triplicates of each protein concentration point were measured and afterwards averaged using Excel software. Graphs were drawn and fitted in SigmaPlot ver. 13.0. The equation used for fitting was a four parameters logistic curve where:

$$y = + \frac{(-min)}{1 + \left(\frac{x}{EC50}\right)^{-Hillslope}},$$

Options were set to default; initial parameters values, as well as parameters min, max, EC50, Hillslope, were selected automatically, parameter constraints were max > min and EC50 > 0, number of iterations was 200, and tolerance was kept at 1e$^{-10}$. The reduced chi-square method was used to compute parameters' standard errors. Experimental r values of respective 0.1 µM BODIPY TMR-lipid and protein-TMR BODIPY-lipid mixtures were compared with theoretical values obtained with the Perrin equation.

## Mass spectrometry analysis of extracted lipids

Lipid extractions from purified recombinant full length and truncated TbMORN1 were achieved by three successive vigorous extractions with ethanol (90% v/v) according to a published protocol [46]. The pooled extracts were dried using N$_2$ gas in a glass vial and re-extracted using a modified Bligh and Dyer method [47]. For whole *E. coli* lipid extracts, cells were washed with PBS and extracted following the modified Bligh and Dyer method. All extracts were dried under N$_2$ gas in glass vials and stored at 4˚C. Extracts were dissolved in 15 µl of CHCl$_3$:CH$_3$OH (1:2) and 15 µl of acetonitrile:propan-2-ol:water (6:7:2) and analysed with an Absceix 4000 QTrap, a triple quadrupole mass spectrometer equipped with a nano-electrospray source. Samples were delivered using a Nanomate interface in direct infusion mode (~125 nl/min). Lipid extracts were analysed in both positive and negative ion modes using a capillary voltage of 1.25 kV. MS/MS scanning (daughter, precursor, and neutral loss scans) were performed using nitrogen as the collision gas with collision energies between 35–90 V, allowing lipid structure assignments.

## Preparation of liposomes and liposome sedimentation assay

The preparation of liposomes has been described previously [48]. Liposomes were prepared from CHCl$_3$ stocks of 10 mg/ml of 1-palmitoyl-2-oleoyl-sn-glycero-3-phosphocholine, POPC, 16:0–18:1 (Avanti Polar Lipids) and 1 mg/ml of porcine brain L-α-phosphatidylinositol 4,5-bisphosphate, PI(4,5)P$_2$ (Avanti Polar Lipids). Lipid mixtures of 0–20% of PI(4,5)P$_2$ and 80–100% of POPC were first dried under a stream of nitrogen, and then vacuum dried for at

least one hour. Dried lipid mixtures were hydrated in gel filtration buffer, incubated for 10 min at RT, gently mixed, further incubated for 20 min, and sonicated for 2 min at RT in a water bath, followed by 5 freeze-thaw cycles. Using a mini-extruder (Avanti Polar Lipids), vesicles were then extruded through polycarbonate filters with 0.1 μm and 0.4 μm pore sizes. TbMORN1(2–15) and Doc2B, a positive control for PI(4,5)P$_2$ binding, were clarified by centrifugation in a TLA-55 rotor for 30 min at 90,720 x g, 4°C, using an Optima MAX-XP tabletop ultracentrifuge (Beckman Coulter Life Sciences), prior to the liposome sedimentation assay. For the assay, proteins and liposomes were mixed in polycarbonate tubes (Beckman Coulter) and incubated for 30 min at RT. The final concentrations of liposomes and proteins were 0.5 mg/ml and 10 μM, respectively. The liposomes were pelleted by centrifugation in an ultracentrifuge for 40 min at 180,743 x g, 20°C, using a TLA-100 rotor (Beckman Coulter). Supernatants and pellets were separated and mixed with 5x and 1x SDS-PAGE loading dye, respectively. Two independent experiments were performed. Samples were separated on 15% acrylamide gels by SDS-PAGE and after electrophoresis, gels were stained with Coomassie Brilliant Blue staining solution and scanned. The amounts of proteins in supernatants and pellets were quantified with QuantiScan 32 software.

## Preparation of Sucrose-Loaded Vesicles (SLVs) and pelleting assay

To generate synthetic SLVs, lipids reconstituted in CHCl$_3$ were mixed in the following ratio: 30% DOPC; 35% DOPE; 15% DOPS; 20% cholesterol. 5 mol % PI(4,5)P$_2$ was added in place of 5 mol % DOPE in the PI(4,5)P$_2$-containing liposomes. Lipids were extracted from bloodstream form *T. brucei* according to an established protocol [49]. Briefly, mid-log phase cells were harvested by centrifugation (750 x g, 10 min, RT), washed once with PBS, then resuspended in 100 μl PBS, and transferred to a glass tube. 375 μl of 1:2 (v/v) CHCl$_3$:MeOH was added and the mixture was vortexed (20 s) and then incubated with continuous agitation (15 min, RT). A further 125 μl CHCl$_3$ was then added to make the mixture biphasic, and following brief vortexing 125 μl ddH$_2$O was added. The mixture was vortexed again and then separated by centrifugation (1000 x g, 5 min, RT). The lower organic layer was then transferred to a new glass vial, dried under a nitrogen stream, and kept at 4°C until use. For the preparation of SLVs from trypanosomal lipids, the lyophilized lipids (extract from 8x10$^7$ cell equivalents) were reconstituted in 50 μl CHCl$_3$. 6 μM Rhodamine B dihexadecanoyl phosphoethanolamine (Rh-DHPE) was added to all lipid mixtures to facilitate the visualisation of the SLVs. The lipid mixtures were dried under a nitrogen stream and the lipid films hydrated in 20 mM HEPES pH 7.4, 0.3 M sucrose. The lipid mixtures were subjected to 4 cycles of freezing in liquid nitrogen followed by thawing in a sonicating water bath at RT. The vesicles were pelleted by centrifugation (250,000 x g, 30 min, RT) and resuspended in 20 mM HEPES pH 7.4, 100 mM KCl to a total lipid concentration of 1 mM. SLVs were incubated with 1.5 μM purified TbMORN1(2–15) in gel filtration buffer (20 mM Tris-HCl pH 8.5, 200 mM NaCl, 2% glycerol, 1 mM DTT) at a 1:1 ratio (30 min, RT). To separate soluble and SLV-bound TbMORN1(2–15), the vesicles were pelleted by centrifugation (8,700 x g, 30 min, RT) and equal volumes of supernatant and resuspended pellet were separated by SDS-PAGE and analysed through Coomassie staining.

## Liposome Microarray Assay (LiMA)

LiMA [50] was performed according to the standard protocol [51]. TbMORN1(2–15) tagged N-terminally with EGFP and two positive controls, PLCδ1-PH and Lactadherin-C2, both fused to superfolder GFP (sfGFP), were applied to microarrays printed with different signalling lipids. In brief, lipids of interest were combined with the carrier lipid DOPC, PEGylated PE, and PE labelled with Atto 647 dye (PE-Atto 647, 0.1 mol%). Lipid mixtures containing 2,

5, and 10 mol% of the signalling lipid were spotted onto a thin agarose layer (TAL). The agarose layers were hydrated using buffer A (20 mM Tris-HCl pH 8.5, 200 mM NaCl) and vesicles formed spontaneously. Efficiency of liposome formation was verified by fluorescence microscopy. The protein was diluted to 7 μM in buffer A and 40 μl was applied to each array. Microarrays were incubated (20 min, RT) and subsequently washed three times with 40 μl of buffer A. Chips were analysed by automated fluorescence microscopy. Positions of liposomes were determined by tracking the fluorescence of PE-Atto 647 and images were taken for 3 ms and 5 ms exposure times. In parallel, the fluorescence of EGFP was determined for 1, 5, 10, 30, 75, 100, 200, and 300 ms exposures. Images were processed using CellProfiler and CPAnalyst. Only EGFP signals that overlapped with Atto 647 signals were taken into account. Normalised binding intensity (NBI) was calculated as the ratio between EGFP and Atto 647 fluorescence, normalised by exposure time. Three microarrays were examined, carrying liposomes with the following signalling lipids; PIP-chip: DOPA, DOPE, DOPI, DOPS, DODAG, cardiolipin, BMP, DOPI(4,5)$P_2$, DOPG; GLP-chip: ceramide C16, ceramide(1)P C16, ceramide(1)P C18, S (1)P, S, SM, DOPI(4,5)$P_2$, DOPS; SL-chip: DOPI(3)P, brain PI(4)P, DOPI(5)P, DOPI(3,4)$P_2$, DOPI(3,5)$P_2$, brain PI(4,5)$P_2$, DOPI(3,4,5)$P_3$, DOPS, and cholesterol. Each microarray was performed in triplicate.

## Cell culture and cell line generation

Bloodstream form *T. brucei* cells were maintained in HMI-9 media supplemented with 10% foetal bovine serum (Sigma-Aldrich, St. Louis, USA) at 37˚C and 5% $CO_2$ in cell culture flasks with filter lids (Greiner). For overexpression studies, 427 strain "single marker" cells—which express T7 RNA polymerase and the Tetracycline repressor protein, both maintained under 2.5 μg/ml G418 selection—were used [52]. For optimisation of digitonin extraction conditions, the GFP[ESPro]-221[ES].121[tet] cell line, which constitutively expresses GFP from the VSG expression site, was used [53]. Constructs for overexpression of Ty1-tagged TbMORN1 and untagged TbMORN1 were obtained by cloning the required ORFs into the pLEW100v5-HYG plasmid; the identity of the inserts was verified by DNA sequencing followed by BLAST analysis against the TbMORN1 ORF (Tb927.6.4670). The plasmids were linearised by NotI digestion, and plasmid DNA was purified by ethanol precipitation. Linearisation was verified using agarose gel electrophoresis. Stable cell lines were generated by using 20 μg of the linearised plasmids to transfect ~3x10[7] "single marker" cells in transfection buffer (90 mM $Na_2PO_4$, 5 mM KCl, 0.15 mM $CaCl_2$, 50 mM HEPES pH 7.3) using the X-001 program of an Amaxa Nucleofector II (Lonza, Switzerland) [54, 55]. Clones were obtained from the transfected cells by limiting dilution under 5 μg/ml hygromycin selection. The presence of the transgene in the genomic DNA of isolated clones was verified by PCR.

## Growth curves and BigEye cell counts

22 ml cells at a defined starting concentration were divided into two flasks of 10 ml each, and overexpression of the ectopic transgene was initiated in one flask by the addition of tetracycline to a final concentration of 1 μg/ml. For overexpression of Ty1-TbMORN1, a starting concentration of 1x10[4] cells/ml was used and the cells were split and reseeded at this concentration after 48 h. For overexpression of untagged TbMORN1, a starting concentration of 1x10[3] cells/ml was used with no reseeding. Tetracycline was refreshed every 24 h in both cases. Population density was measured using a Z2 Coulter Counter (Beckman Coulter, Krefeld, Germany) at the indicated timepoints. For quantification of BigEye cell incidence at the indicated timepoints, the cultures were briefly agitated to mix the cells, which were then allowed to settle for 30 min. The culture flasks were then examined directly using an inverted

phase contrast microscope (Leitz Labovert) and a 10x objective lens. Three fields of view were chosen at random for each flask and the number of normal and BigEye cells manually quantified, using higher magnification where necessary. Given the low magnification used, the numbers presented are likely to be underestimates of the true incidence.

## Immunoblotting

To obtain whole cell lysates, cell concentration was measured using a Z2 Coulter Counter, and a defined volume was then transferred to 15 ml Falcon tubes. The cells were pelleted by centrifugation (750 x g, 10 min, RT), resuspended in 1 ml PBS, and transferred to microfuge tubes. The cells were again pelleted (1800 x g, 2 min, RT) and the cell pellet then directly resuspended in SDS loading buffer to a final concentration of $2 \times 10^5$ cells/μl. The lysates were heated (95˚C, 10 min) before use. Lysates were separated by SDS-PAGE ($1.4 \times 10^6$ cells/lane in a 15-well gel of 1.0 mm thickness) and the proteins then transferred to nitrocellulose membranes. The membranes were blocked in blocking buffer (10% milk, PBS, 0.3% TWEEN-20) (30 min, RT) and then incubated with the indicated primary antibodies in blocking buffer (1 h, RT). The membranes were washed three times in PBS-T (PBS, 0.3% TWEEN-20) and were then incubated with IRDye-conjugated secondary antibodies in PBS-T (1 h, RT). After a further three washes in PBS-T the membranes were briefly dried between sheets of filter paper and then imaged using an Odyssey CLx (LI-COR Biosciences, Bad Homberg, Germany). Processing and quantification was carried out using ImageStudioLite software (LI-COR Biosciences).

## Immunofluorescence microscopy

Cell concentration was measured using a Z2 Coulter Counter and $10^6$ cells per coverslip were taken. The cells were transferred to 15 ml Falcon tubes and fixed directly in media by the addition of paraformaldehyde solution to a final concentration of 4% (37˚C, 20 min). 10 ml trypanosome dilution buffer (TDB; 20 mM $Na_2HPO_4$, 2 mM $NaH_2PO_4$, 5 mM KCl, 80 mM NaCl, 1 mM $MgSO_4$, 20 mM glucose) was then added and the cells were pelleted by centrifugation (750 x g, 10 min, RT). The supernatant was removed, the cell pellet was resuspended in 500 μl TDB, and the cells were transferred to poly-L-lysine-coated coverslips in a 24-well plate. The cells were attached to the coverslips by centrifugation (750 x g, 4 min, RT) and the cells were permeabilised using a solution of 0.25% TritonX-100 in PBS (5 min, RT). The cells were washed with PBS, blocked using a solution of 3% BSA in PBS (30 min, RT), and sequentially incubated with primary and secondary antibodies diluted in PBS (1 h, RT for each) with three PBS wash steps after each incubation. After the final wash, the coverslips were rinsed in ddH$_2$O, excess fluid removed by wicking, and mounted on glass slides using Fluoromount-DAPI (Southern Biotech). For analysis of detergent-extracted cytoskeletons, cells were washed using TDB and attached to poly-L-lysine coverslips as described above. The cells were incubated in extraction buffer (0.5% IGEPAL, 0.1 M PIPES-NaOH pH 6.9, 2 mM EGTA, 1 mM $MgSO_4$, 0.1 mM EDTA, cOmplete protease inhibitors [Roche]) (5 min, RT), washed three times with extraction buffer, and then fixed with ice-cold MeOH (-20˚C, 30 min). Blocking, antibody incubation steps, and mounting were as described above. All liquid handling was carried out using a P1000 micropipette, and pipetting was done as gently as possible to minimise shear forces. The coverslips were imaged using a Leica DMI6000B inverted microscope equipped with a Leica DFC365 camera and a 100x oil objective lens (NA1.4) and running Leica Application Suite X software. The same exposure times were used for acquisition of +/-Tet samples, and 40 z-slices of 0.21 μm thickness were taken per field of view. Image processing was carried out using ImageJ. Maximum intensity z-projections are shown.

## Fractionation

Cell concentration was measured using a Z2 Coulter Counter and an equal number of cells ($\sim 2.5 \times 10^7$ per experiment) was taken from the control (-Tet) and overexpression (+Tet) samples and transferred to 50 ml Falcon tubes. The cells were pelleted by centrifugation (750 x g, 10 min, 4˚C) and the cell pellets then resuspended in 1 ml TDB and transferred to microfuge tubes. The cells were pelleted by centrifugation (1800 x g, 2 min, 4˚C) and then resuspended in 200 µl extraction buffer (see Immunofluorescence microscopy section for composition). After a short incubation (15 min, RT, orbital mixer), a 5% (10 µl) input sample was taken and the mixtures separated by centrifugation (3400 x g, 2 min, 4˚C) into detergent-soluble (cytoplasmic) supernatant and detergent-insoluble (cytoskeleton) pellet fractions. The supernatant was transferred to a fresh microfuge tube, its exact volume noted, and a 5% sample taken. The tube containing the pellet was centrifuged a second time (3400 x g, 2 min, 4˚C) in order to bring down material sticking to the tube wall; this second supernatant was discarded. The pellet was resuspended in 200 µl extraction buffer and a 5% sample (10 µl) taken. SDS loading buffer was added to the input, supernatant, and pellet samples to a final volume of 20 µl and denaturation assisted by heating (95˚C, 10 min). Equal fractions were loaded onto polyacrylamide gels, separated by SDS-PAGE, and analysed by immunoblotting. In the exemplary blot shown (Fig 5G), each sample is a 4.5% fraction, equivalent to $\sim 10^6$ cells in the Input fraction. For optimisation of extraction conditions using digitonin, essentially the same protocol was followed except that ultra-pure digitonin (Calbiochem) in TDB buffer was used and incubations were carried out at 24˚C in a heating block. For the two-step digitonin/IGEPAL fractionations (S8 Fig), cells were pelleted by centrifugation (750 x g, 10 min, RT), resuspended in 1 ml TDB, transferred to microfuge tubes, and pelleted again (750 x g, 3 min, RT). The cell pellet was resuspended in 400 µl of 40 µg/ml digitonin in TDB and extracted (25 min, 24˚C), after which a 5% input sample was taken. The mixture was then separated by centrifugation (750 x g, 5 min, RT) and 320 µl of the cytosolic fraction (SN1) transferred to a fresh tube and a 5% sample was taken. The cell pellet was then resuspended with 1 ml TDB and the extracted cells again pelleted by centrifugation (750 x g, 5 min, RT). The extracted cells were then resuspended in 400 µl extraction buffer (see Immunofluorescence section above for composition) and incubated (24˚C, 15 min, heating block with shaker). After the incubation, a 5% sample (P1) was taken and the extracted cells pelleted by centrifugation (3400 x g, 2 min, RT). 320 µl of the supernatant (SN2) was transferred to a fresh microfuge tube and a 5% sample was taken. The pellet was resuspended in 1 ml TDB and centrifuged again (750 x g, 5 min, RT). The pellet (P2) was then resuspended in 400 µl extraction buffer. SDS loading buffer was added to the 5% samples (I, SN1, SN2, P1, P2) to a final volume of 40 µl. Samples were analysed by immunoblotting as detailed above.

## Crystallisation

Crystallisation of TbMORN1(7–15) (with a C-terminal Strep tag), TgMORN1(7–15) and PfMORN1(7–15) (both with affinity tags removed) was performed at 22˚C using a sitting-drop vapour diffusion technique and micro-dispensing liquid handling robots (Phoenix RE (Art Robbins Instruments) and Mosquito (TTP labtech)). In the case of TbMORN1(7–15), crystals only appeared from reductively methylated protein, using a standard protocol (Walter et al., 2006). The best diffracting crystals of TbMORN1(7–15) were grown at a protein concentration of 3.5 mg/ml in the following conditions: 0.166 M Tris-HCl pH 8.8, 0.15 M MgCl₂, 0.45 M KI, 24% PEG 2000 MME, and 4% glycerol. The tetragonal crystals of TgMORN1(7–15) were obtained at a protein concentration of 10 mg/ml in the following conditions: 0.1 M Tris-HCl pH 8.2, 15% PEG 3350, 0.2 M NaCl. The diffracting crystals of both selenomethionine-

containing crystals and native crystals of PfMORN1(7–15) were obtained at a protein concentration of 8 mg/ml in the conditions "B11" from the Morpheus II crystallisation screen (Molecular Dimensions): 2 mM divalents mix (0.5 mM MnCl₃, 0.5 mM CoCl₂, 0.5 mM NiCl₂, 0.5 mM Zn(OAc)₂, 0.1 M Buffer System 6, pH 8.5 (Gly-Gly, AMPD), and 50% precipitation Mix 7 (20% PEG 8000, 40% 1,5-Pentanediol). The crystals were flash cooled in liquid nitrogen prior to data collection.

## X-ray diffraction data collection and crystal structure determination

Initially, the structure of PfMORN1(7–15) was determined using the single-wavelength anomalous diffraction (SAD) method. The selenomethionine dataset was collected at the beamline ID29 (ESRF, Grenoble) at 100K at the peak of selenium using a wavelength of 0.979 Å. The data frames were processed using the XDS package [56], and converted to mtz format with the program AIMLESS [57]. The apo-PfMORN1(7–15) structure was solved using single anomalous diffraction with AUTOSOL software from the PHENIX package. The structures of TgMORN1(7–15) and TbMORN1(7–15) were then solved using the molecular replacement program PHASER [58] with the atomic coordinates of PfMORN1(7–15) as a search model. The structures were then refined with REFMAC and Phenix Refine and rebuilt using Coot [59–61]. The structures were validated and corrected using the PDB_REDO server [62]. The figures were produced using PyMOL (Schrödinger, Inc.) and Chimera software [63]. Coordinates have been deposited in the protein data bank (accession codes 6T4D, 6T4R, 6T68, 6T69, 6T6Q). Data collection and refinement statistics are reported in Table 1.

## Small Angle X-ray Scattering (SAXS)

Synchrotron radiation X-ray scattering from various MORN constructs in solution were collected at different synchrotron facilities (S3 Table). TgMORN1(7–15), TbMORN1(7–15) as well as PfMORN1(7–15) were collected at the EMBL P12 beamline of the storage ring PETRA III (DESY, Hamburg, Germany) [64]. Images were recorded using a photon counting Pilatus-2M detector at a sample to detector distance of 3.1 m and a wavelength ($\lambda$) of 1.2 Å covering the range of momentum transfer $0.01 < s < 0.5$ Å$^{-1}$; with $s = 4\pi\sin\theta/\lambda$, where $2\theta$ is the scattering angle. To obtain data from a monodisperse sample from TgMORN1(7–15) and TbMORN1(7–15), a size exclusion chromatography column directly coupled to the scattering experiment (SEC-SAXS) was employed. The parallel collection of UV and light scattering data allowed the protein to be monitored while it eluted from the column [65]. Throughout the complete chromatography process, 1 s sample exposures were recorded. Various buffers were used as the mobile phase; TbMORN1(7–15): 20mM Tris-HCl pH 8.5, 200 mM NaCl, 2% (v/v) glycerol, 1 mM DTT; TgMORN1(7–15): 20 mM Tris-HCl pH 7.5, 100 mM NaCl. 100 µl of purified sample (3.8 mg/mL TbMORN1(7–15) and 2.6 mg/ml TgMORN1(7–15)) were injected onto a Superdex 200 10/300 (GE Healthcare) column and the flow rate was set to 0.5 ml/min. SAXS data were recorded from macromolecular free fractions corresponding to the matched solvent blank. PfMORN1(7–15) was measured in batch mode from a concentration series spanning 1–8 mg/ml. 20 mM Tris-HCl pH 7.5, 100 mM NaCl buffer was measured for background subtraction. As a concentration dependent increase in size was detectable further analysis were based solely on the data collected at 1 mg/ml. In a similar manner as described above, TbMORN1(2–15) data were collected at ESRF BM29 beamline [66] in SEC-SAXS mode with the setup described by Brennich et al. [67]. SAXS data from the run were collected at a wavelength of 0.99 Å using a sample-to-detector (PILATUS 1 M, DECTRIS) distance of 2.867 m. Here too, a Superdex 200 10/300 (GE Healthcare) column was used as well as 20mM Tris-HCl pH 8.5, 200 mM NaCl, 2% (v/v) glycerol, 1 mM DTT as mobile phase. 100 µl

**Table 1. X-ray data collection statistics.**

| | PfMORN (SMet) | PfMORN | TbMORN | TbMORN | TgMORN | TgMORN (V-shape) |
|---|---|---|---|---|---|---|
| Source | ID29 | ID29 | ESRF ID23-1 | ESRF ID23-1 | ESRF ID30B | ESRF ID30B |
| Wavelength (Å) | 0.979 | 0.976 | 0.979 | 1.89 | 0.967 | 1.0 |
| Resolution (Å) | 47.47–2.5 | 46.3–2.14 | 48.28–2.35 | 48.14–2.53 | 48.92–2.90 | 49.45–2.50 |
| | (2.59–2.5) | (2.2–2.14) | (2.43–2.35) | (2.65–2.53) | (3.08–2.90) | (2.60–2.50) |
| Space group | $C222_1$ | $C222_1$ | $P2_1$ | $C2$ | $P4_32_12$ | $P6_222$ |
| Unit cell (Å, °) | a = 57.33, b = 79.18, c = 94.42 | a = 57.33, b = 79.18, c = 94.42 | a = 69.04, b = 27.63, c = 114.54; β = 101.83 | a = 192.88, b = 49.74, c = 41.98 | a = b = 53.92c = 348.85 | a = b = 205.86, c = 40.58 |
| Molecules (a.u.) | 1 | 1 | 2 | 2 | 2 | 1 |
| Unique reflections | 7832 | 12148 | 17460 | 12419 | 12331 | 18162 |
| | (762) | (974) | (1247) | (975) | (1819) | (1960) |
| Completeness (%) | 99.5 | 99.6 | 95.4 | 93.1 | 99.1 | 99.7 |
| | (98.3) | (98.1) | (71.5) | (60.0) | (95.4) | (97.9) |
| $R_{merge}$ [b] | 0.059 | 0.037 | 0.092 | 0.116 | 0.140 | 0.198 |
| | (0.301) | (1.283) | (0.466) | (1.365) | (2.252) | (2.116) |
| $R_{meas}$ [c] | 0.062 | 0.042 | 0.100 | 0.146 | 0.158 | 0.204 |
| | (0.314) | (1.510) | (0.532) | (1.893) | (2.536) | (2.177) |
| CC(1/2) | 0.999 | 1.000 | 0.998 | 0.995 | 0.999 | 0.999 |
| | (0.983) | (0.467) | (0.897) | (0.303) | (0.772) | (0.426) |
| Multiplicity | 13.0 (12.9) | 4.2 (3.2) | 6.1 (4.1) | 6.2 (3.7) | 8.4 (8.0) | 17.9 (18.1) |
| $I$/sig($I$) | 30.3 (8.1) | 17.5 (0.8) | 13.4 (2.8) | 9.3 (0.9) | 11.8 (0.5) | 14.6 (1.6) |
| $B_{Wilson}$ (Å$^2$) | 57.3 | 58.0 | 18.74 | 36.7 | 22.97 | 42.3 |
| $R_{work}$ [d]/$R_{free}$ [e] | | 23.0/26.4 | 23.2/25.6 | 22.5/28.2 | 31.8/33.8 | 20.1/23.9 |
| r.m.s.d. bonds (Å) | | 0.003 | 0.004 | 0.011 | 0.0084 | 0.007 |
| r.m.s.d. angles (°) | | 0.6 | 1.24 | 1.612 | 1.417 | 0.811 |

[a] Values in parentheses are for the highest resolution shell.

[b] $$R_{merge} = \frac{\sum_{hkl}\sum_{i=1}^{N}|I_{i(hkl)} - \bar{I}_{(hkl)}|}{\sum_{hkl}\sum_{i=1}^{N}I_{i(hkl)}}$$

[c] $$R_{meas} = \frac{\sum_{hkl}\sqrt{N/(N-1)}\sum_{i=1}^{N}|I_{i(hkl)} - \bar{I}_{(hkl)}|}{\sum_{hkl}\sum_{i=1}^{N}I_{i(hkl)}}$$

where $\bar{I}_{(hkl)}$ is the mean intensity of multiple $I_{i(hkl)}$ observations of the symmetry-related reflections and N is the redundancy

[d] $$R_{work} = \frac{\sum||F_{obs}|-|F_{calc}||}{\sum|F_{obs}|}$$

[e] $R_{free}$ is the cross-validation $R_{factor}$ computed for the test set of reflections (5%) which are omitted in the refinement process.

of 5.8 mg/ml TbMORN1(2–15) were injected. Data reduction to produce final scattering profiles of dimeric MORN1 constructs were performed using standard methods. Briefly, 2D-to-1D radial averaging was performed using the SASFLOW pipeline (Franke et al., 2017). For data collected at ESRF, EDNA pipeline [68] was used. CHROMIXS was used for the visualisation and reduction of the SEC-SAXS datasets [69]. Aided by the integrated prediction algorithms in CHROMIXS, the optimal frames within the elution peak and the buffer regions were

selected. Single buffer frames were then subtracted from sample frames one by one, scaled, and averaged to produce the final subtracted curve. The indirect inverse Fourier transform of the SAXS data and the corresponding probable real space-scattering pair distance distribution (p(r) versus r profile) of the various MORN1 constructs were calculated using GNOM [70], from which the $R_g$ and $D_{max}$ were determined. The p(r) versus r profile were also used for ab initio bead modelling of selected MORN1 constructs. For this, 20 independent runs of DAM-MIF [71] in the case of TbMORN1(2–15) and DAMMIN [72] in case of the shorter MORN1 (7–15) constructs were performed. From these the most probable models were selected by DAMAVER [73]. The ab initio modelling was performed with and without symmetry constraints (p2 symmetry to reflect the dimeric state of the protein). Comparison with theoretical curves calculated from the X-tal structures described here was performed with Crysol [74]. Due to the elongated nature of the molecules, fits were improved by increasing LM (maximum order of harmonics) to 50. The molecular mass (MM) was evaluated based on concentration independent methods according to Porod [75] and as implemented in the ATSAS package. Dimensionless Kratky plots were constructed according to [76] and the reference point for globular proteins at $\sqrt{3}$, 1.1 indicated. Graphical representation was produced using PyMOL Molecular Graphics System (Schrödinger, Inc.). The SAXS data (S3 Fig) and ab initio bead models as well as fits to the crystal structures described within this work have been deposited into the Small-Angle Scattering Biological Data Bank (SASBDB) [77] under the accession codes SASDG97, SASDGA7, SASDGB7, SASDGC7.

### Transmission Electron Microscopy (EM) with rotary shadowing

TbMORN1 and TbMORN1(2–15) were purified according to the two-step procedure detailed above. They were then diluted in spraying buffer (100 mM $NH_4CH_3CO_2$-NaOH pH 8.5, 30% (v/v) glycerol) to a final concentration of 50–100 μg/ml. Diluted samples were sprayed onto freshly cleaved mica chips (Christine Gröpl) and immediately transferred into a MED020 high vacuum evaporator (BAL-TEC) equipped with electron guns. While rotating, samples were coated with 0.6 nm of Platinum (BALTIC) at an angle of 4˚, followed by 6 nm of Carbon (Oerlicon) at 90˚. The obtained replicas were floated off the mica chips, transferred to 400 mesh Cu/Pd grids (Agar Scientific), and examined using a Morgagni 268D transmission electron microscope (FEI) operated at 80 kV. Images were acquired using an 11 megapixel Morada CCD camera (Olympus-SIS).

## Results

### TbMORN1 forms tail-to-tail dimers via its C-terminus

TbMORN1 is composed of 15 consecutive 23-amino acid MORN repeats with a 5-amino acid extension after the sixth repeat (Fig 1A). An alignment of the repeats in TbMORN1 revealed several highly conserved glycine residues, with a rough consensus of YxGEWx$_2$Gx$_3$GxG-x$_3$Yx$_2$Gx$_2$ (Fig 1A, sequence logo). Bioinformatic analysis of TbMORN1 predicted an all-β secondary structure, with each repeat expected to form a β-hairpin (strand-loop-strand) pattern (Fig 1A).

Limited proteolysis showed progressive digestion occurring from the N-terminus, while the C-terminus was fairly stable (S1A and S1B Fig). Consequently, a panel of different truncations was cloned according to the MORN repeat boundaries predicted by the alignment (Fig 1A). These truncations were named according to the number of repeats (1–15) they contained. These constructs were expressed in bacteria and purified using a two-step protocol combining affinity purification and size exclusion chromatography (S1C Fig).

The yields of TbMORN1(1–15) were always very low due to aggregation in the first purification step, making this construct not generally suitable for molecular biophysics assays. Size exclusion chromatography coupled to multi-angle light scattering (SEC-MALS) analysis of purified TbMORN1(1–15) elution profiles suggested the formation of higher-order assemblies (Fig 1B). By contrast, TbMORN1(2–15) displayed a well-defined monodisperse peak in SEC-MALS, with a molecular weight corresponding to a dimer (Fig 1B). This strongly suggested that the first MORN repeat mediated oligomerisation. Further successive truncations from the N-terminus (TbMORN1(7–15) and TbMORN1(10–15)) also eluted as monodisperse dimers, suggesting that dimerisation of TbMORN1 is mediated by the C-terminus (S1D Fig). Consistent with this conclusion, removal of the last MORN repeat in the construct TbMORN1(1–14) resulted in the elution of a mixture of monomers, dimers, and higher-order structures (Fig 1B). This demonstrated that the C-terminal repeats play an important role in dimer stabilisation.

Circular dichroism (CD) measurements taken of TbMORN1(1–15), TbMORN1(2–15), and TbMORN1(7–15) indicated >30% antiparallel β-strand content for each construct (S1E Fig) and were in good agreement with a priori bioinformatic predictions (Fig 1A). Thermostability measurements of TbMORN1(1–15), TbMORN1(2–15) and TbMORN1(7–15) using CD returned values in ˚C of 45.6 +/- 0.1, 43.5+/-0.1, and 42.2+/-0.1 respectively, showing an expected incremental contribution of MORN repeats to the overall stability of the protein.

To map which residues were likely mediating dimerisation, cross-linking mass spectrometry (XL-MS) was used to analyse TbMORN1(1–15). Two different chemical cross-linkers were used: EDC (1-ethyl-3-(3-dimethylaminopropyl)carbodiimide), which has a zero-length spacer arm, and BS³ (bis(sulphosuccinimidyl)suberate), which has an 11.4 Å spacer arm. For both chemicals, most cross-links were observed to form between repeats 13, 14, and 15, especially via repeat 14 (S1 Table). These data were consistent with those obtained by SEC-MALS (Figs 1B and S1D), suggesting that TbMORN1 forms tail-to-tail dimers via their C-termini. It remained unclear whether the polypeptide chains in dimers were in a parallel or antiparallel orientation, however.

## TbMORN1 can bind lipid side chains but not phospholipid liposomes

To identify candidate phospholipids that interacted with TbMORN1, we initially used PIP strips. These resulted in the identification of a number of candidates, of which PI(4,5)P$_2$ was the most interesting due to its well-established role as an endocytic cofactor (S2A and S2B Fig). We subsequently used fluorescence anisotropy (FA) to quantify binding of TbMORN1 to PI(4,5)P$_2$, and found that all three N-terminal truncations of TbMORN1 bound to PI(4,5)P$_2$ with micromolar affinity (S2C Fig). This indicated that the C-terminus of TbMORN1 was mediating binding to PI(4,5)P$_2$. Binding of TbMORN1(2–15) and TbMORN1(10–15) was additionally validated in native gel bandshift assays (S3A and S3B Fig). Unexpectedly, these results could not subsequently be validated in liposome binding assays (S2D and S2E Fig). We carried out additional controls modifying liposome curvatures and concentrating PI(4,5)P$_2$ in microdomains by addition of cholesterol but this did not result in binding (S4A Fig). Through sequence analysis we identified putative binding sites in repeats 13 and 14 and mutagenised selected Arg and Lys residues to Ala. Surprisingly, when both sites were mutated simultaneously, the protein became monomeric (S4B Fig). The mutagenesis did not however affect secondary structure content (S4C Fig) or lipid binding as assessed by FA (S4D Fig). Subsequent FA assays showed that the positive results of the FA experiments were principally due to an interaction with longer aliphatic chains rather than the lipid headgroup (S5 Fig).

## TbMORN1 co-purifies with PE but does not bind to lipid vesicles in vitro

A caveat to the previous conclusion was that the purified recombinant TbMORN1 was obtained via mechanical lysis of bacterial cells in the absence of detergent (see Materials and Methods). It was therefore possible that bacterial lipids might be occupying the lipid binding sites on TbMORN1.

To test this, purified recombinant TbMORN1(1–15) and TbMORN1(10–15) were treated according to a de-lipidation protocol and the resulting supernatants were analysed by mass spectrometry. Interestingly, large amounts of phosphatidylethanolamine (PE) were found to have been bound to TbMORN1(1–15) (Fig 2A). The co-purifying PE displayed a remarkably

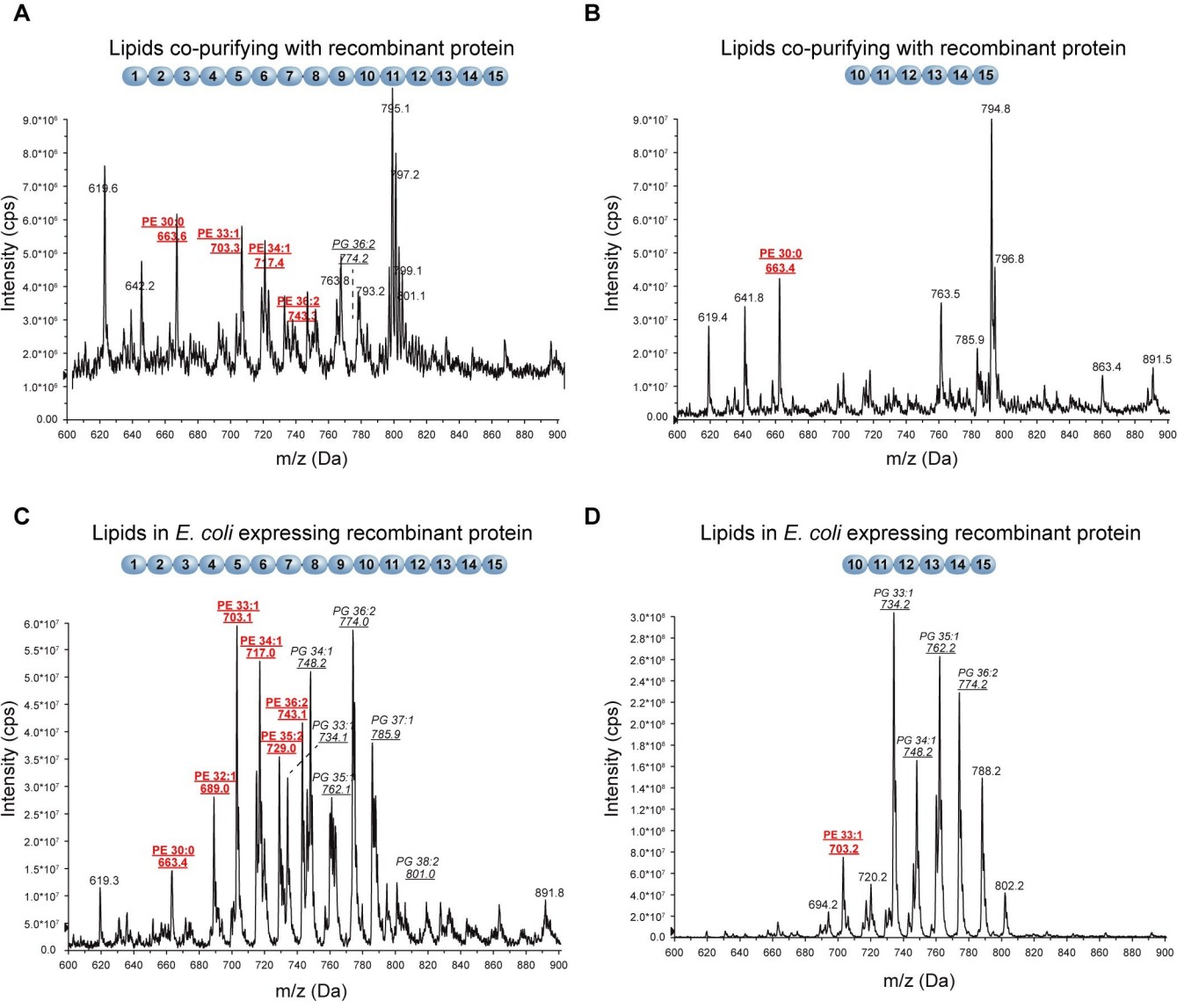

**Fig 2. Recombinant TbMORN1 co-purifies with PE and increases _E. coli_ PE levels.** Negative ion mode survey scan (600–900 m/z) of lipid extracts from the indicated conditions. **(A,B)** Lipid extracts from purified recombinant TbMORN1(1–15) **(A)** and TbMORN1(10–15) **(B)**. A large amount of PE co-purified with TbMORN1(1–15) but very little was associated with TbMORN1(10–15). **(C,D)** Lipid extracts from _E. coli_ cells expressing the indicated constructs. **(C)** Cells expressing TbMORN1(1–15) had elevated PE levels. **(D)** Cells expressing TbMORN1(10–15) showed no changes to cellular lipid ratios compared to wild-type (empty vector control). In all cases, phospholipid identity was confirmed by daughter fragmentation and reported here. Schematics of the recombinant TbMORN1 constructs are shown above the traces.

narrow window of molecular moieties differing in the length of the aliphatic chain (30:0, 33:1, 34:1, 36:2). No significant PE presence was detected in supernatants obtained following de-lipidation of TbMORN1(10–15) (Fig 2B). Given that TbMORN1(10–15) showed robust binding to PI(4,5)P$_2$ with a longer 16-carbon aliphatic chain (S2C Fig), this suggested that PE was not occluding the binding site.

To remove co-purifying lipids, recombinant TbMORN1(1–15) was purified using hydrophobic interaction chromatography—in this regime, almost no lipids were detected in the elutions. Triton X-100 treatment was also found to efficiently remove bound lipids.

Co-purification of PE with TbMORN1(1–15) is not evidence of physiological interaction, as PE is highly abundant in bacteria and carries a net neutral charge [78]. As such, PE might simply have associated with the recombinant protein following lysis of the bacteria prior to purification. To investigate this, mass spectrometry analysis of whole-cell lipid extracts from bacteria expressing recombinant TbMORN1(1–15) or TbMORN1(10–15) was carried out. Curiously, bacterial cells expressing TbMORN1(1–15) showed elevated levels of PE (Fig 2C). This effect was not seen in bacterial cells expressing TbMORN1(10–15), which had approximately wild-type levels of PE (Fig 2D).

In summary, TbMORN1(1–15) but not TbMORN1(10–15) was found to have co-purifying PE and bacterial cells expressing TbMORN1(1–15) appeared to have elevated levels of PE. This suggested that PE might be a plausible candidate for a physiological interaction partner of TbMORN1 and that it requires MORN repeats 1–9 for binding.

To test the possibility of PE binding and also to check whether the presence of co-purifying PE was affecting possible PI(4,5)P$_2$ binding, pelleting assays using sucrose-loaded vesicles (SLVs) were carried out. No association with liposomes was seen regardless if the protein was treated with Triton X-100 beforehand (S6A–S6C Fig).

Finally, in an unbiased and high-throughput approach, TbMORN1 was tested in a liposome microarray assay (LiMA). The EGFP-TbMORN1(2–15) protein showed no significant preference or affinity for liposomes across the whole range of conditions tested (S6D and S6E Fig).

Taken together, these data suggest that TbMORN1 requires long aliphatic chains of the lipid for binding, which would explain the negative results in the liposome-based assays.

## TbMORN1 does not associate with membranes in vivo

At this point, the only remaining positive indicators of an interaction of TbMORN1 with lipids came from the PIP strips (S2A Fig), and the bacterial mass spectrometry data (S2A and S2C Fig). Both these positive pieces of evidence related to TbMORN1(1–15), which was polydisperse in vitro and formed large oligomers (Fig 1B). It was not possible to test the membrane-binding activity of these polydisperse oligomers in vitro however, as the purification yields of TbMORN1(1–15) were always low. As an alternative, the possible membrane association of TbMORN1(1–15) was examined in vivo.

For these experiments, cell lines of bloodstream form *T. brucei* were generated that inducibly expressed TbMORN1(1–15) with an N-terminal Ty1 epitope tag. Induction of Ty1-Tb-MORN1(1–15) overexpression using tetracycline (Tet) produced a strong growth defect in all three *T. brucei* clones tested (Fig 3A). A rise in the number of so-called "BigEye" cells with grossly enlarged flagellar pockets was seen in the same 96-hour time window (Fig 3B). Such a phenotype had previously been seen following depletion of TbMORN1 [25]. The BigEye phenotype is thought to result from perturbations to membrane traffic, especially endocytosis [79].

Immunoblotting with anti-TbMORN1 antibodies confirmed tight and inducible expression of the ectopic Ty1-TbMORN1 protein (Fig 3C, left panel). The presence of the Ty1 epitope tag

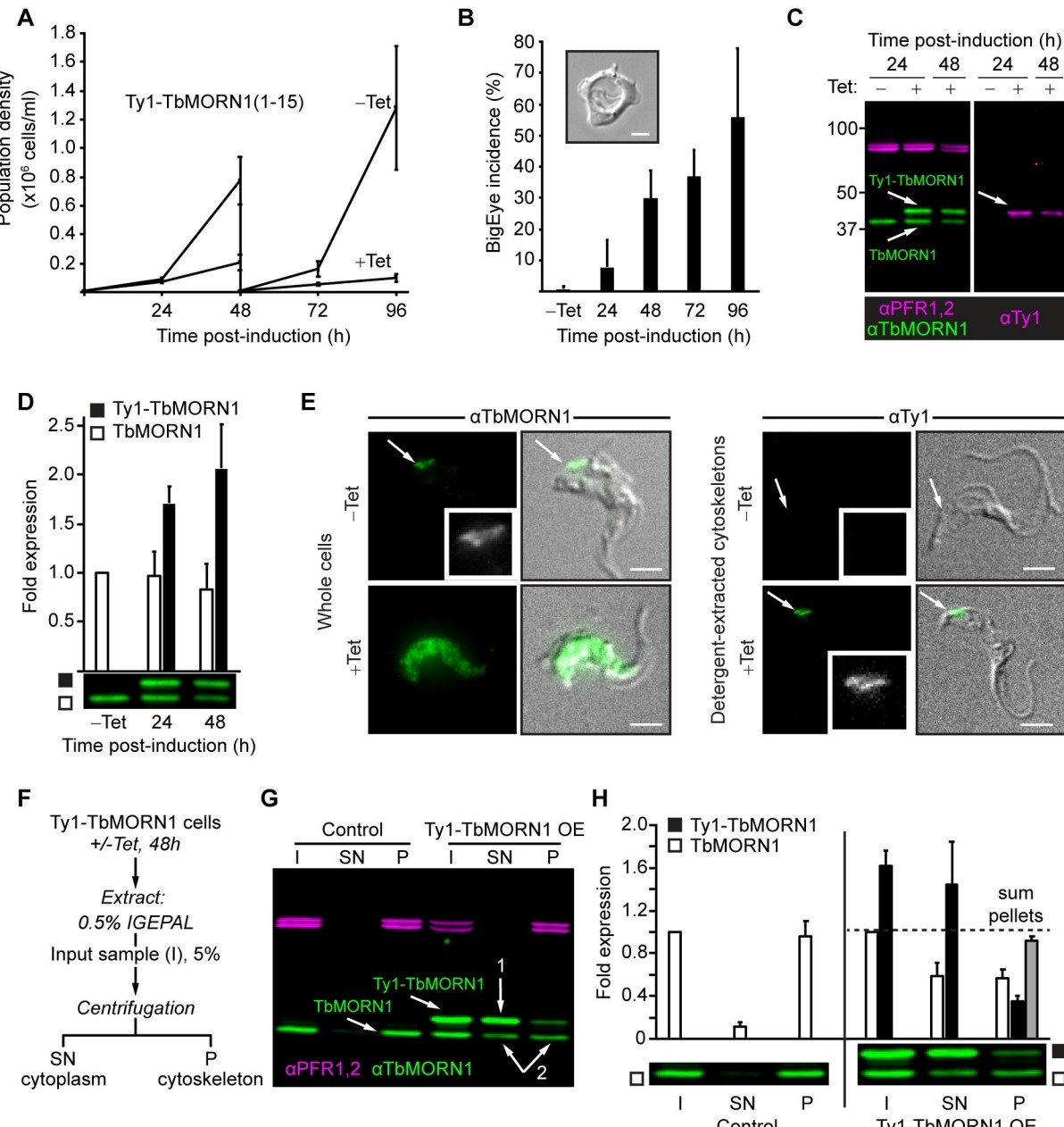

**Fig 3. Overexpression of Ty1-tagged TbMORN1 causes a dominant negative phenotype. (A)** Overexpression of Ty1-TbMORN1 is deleterious. Growth curves of control (-Tet) cells, and cells inducibly expressing Ty1-TbMORN1(1–15) (+Tet). Population density was measured every 24h, and the cultures split and reseeded at 48h. Data were compiled from 3 separate clones, each induced in 3 independent experiments; bars show mean +/- SD. **(B)** Overexpression of Ty1-TbMORN1 produces a BigEye phenotype. The incidence of BigEye cells was counted in control (-Tet) and Ty1-TbMORN1-expressing cells at the indicated timepoints. Data were compiled from 3 separate clones, each induced in 3 independent experiments; bars show mean + SD. The inset shows an example BigEye cell. Scale bar, 2 μm. **(C)** Tight induction of Ty1-TbMORN1 expression. Whole-cell lysates were harvested from control (-Tet) and Ty1-TbMORN1-expressing cells (+Tet) at the indicated timepoints and probed with anti-TbMORN1 and anti-Ty1 antibodies. PFR1,2 were used as a loading control. At least three independent inductions were carried out for each clone; an exemplary blot is shown. **(D)** Quantification of overexpression. The levels of endogenous TbMORN1 and ectopic Ty1-TbMORN1 in immunoblots were normalised relative to the PFR1,2 signal. Data were compiled using 3 separate clones, each induced in at least two independent experiments; bars show mean + SD. **(E)** Ty1-TbMORN1 can localise correctly to the cytoskeleton. Whole cells or detergent-extracted cytoskeletons were fixed and labelled with anti-TbMORN1 or anti-Ty1 antibodies. The fluorescence signal is shown with the transmitted light image of the cell overlaid; inset shows the fluorescence signal from the antibody in greyscale. Results confirmed for 3 separate clones, exemplary images are shown. Scale bars, 2 μm. **(F)** Schematic of the fractionation protocol. **(G)** Overexpression of Ty1-TbMORN1 displaces the endogenous protein from the cytoskeleton. Control and Ty1-TbMORN1-expressing cells were fractionated as shown in panel F and the I, SN, and P fractions were blotted. PFR1,2 was used as a

marker for the cytoskeleton. Expression of Ty1-TbMORN1 was accompanied by a displacement of endogenous TbMORN1 from the insoluble (P) fraction into the soluble (SN) fraction (arrows 1,2). Equal fractions (5%) were loaded in each lane. 3 independent experiments using 3 separate clones were carried out; an exemplary blot is shown. **(H)** Quantification of the fractionation data. Bars show mean + SD.

in the ectopic protein was confirmed by blotting with anti-Ty1 antibodies (Fig 3C, right panel). Quantification of the immunoblots indicated only around 2-fold overexpression of Ty1-TbMORN1 relative to endogenous TbMORN1 protein (Fig 3D).

The strong growth defect and the production of BigEye cells were unexpected and could potentially be due either to the overexpression of the TbMORN1 protein or the presence of the Ty1 tag. To check whether overexpression alone could drive a negative outcome, cells that inducibly expressed untagged TbMORN1 from an ectopic locus were generated. Overexpression of untagged TbMORN1 from the ectopic locus was detected by immunoblotting (S7A Fig). Quantification of overexpression indicated that approximately 7 times more TbMORN1 protein was present in induced cells relative to controls (S7B Fig). Cells overexpressing untagged TbMORN1 exhibited a very strong growth defect and the BigEye phenotype, stronger than that seen for overexpression of Ty1-TbMORN1 (S7C Fig). Therefore, TbMORN1 protein levels alone were capable of producing a negative effect on growth in the absence of the Ty1 tag. It was however not possible to obtain cells that solely expressed Ty1-TbMORN1 by endogenous replacement, despite repeated attempts. This indicated that Ty1-TbMORN1 cannot functionally compensate for the loss of the untagged endogenous protein.

The localisation of the overexpressed Ty1-TbMORN1 was assessed using immunofluorescence microscopy. Uninduced cells labelled with anti-TbMORN1 antibodies showed the expected localisation of endogenous TbMORN1 at the hook complex. Conversely, the Ty1-TbMORN1-overexpressing cells displayed a whole-cell labelling pattern (Fig 3E, left panels). In these cells no sign of protein aggregation was observed, but the different distribution compared to the controls suggested that the Ty1-TbMORN1 protein might not localise correctly. Immunofluorescence microscopy analysis of detergent-extracted cytoskeleton fractions labelled using anti-Ty1 antibodies confirmed that Ty1-TbMORN1 was able to target correctly to the hook complex; as expected, no labelling was seen in uninduced controls (Fig 3E, right panels).

To confirm the immunofluorescence microscopy observations, one-step biochemical fractionation using the non-ionic detergent IGEPAL was used. The detergent-soluble cytoplasmic fraction (supernatant, SN) was separated from the detergent-insoluble cytoskeleton (pellet, P) by centrifugation (Fig 3F). In control cells, endogenous TbMORN1 associated almost entirely with the cytoskeletal (P) fraction (Fig 3G). Blotting fractions from Ty1-TbMORN1-overexpressing cells with anti-TbMORN1 antibodies showed that the overexpressed Ty1-TbMORN1 was mostly detergent-soluble and therefore present in the cytoplasmic fraction (Fig 3G, arrow 1). However, a small amount of the Ty1-TbMORN1 protein associated with the cytoskeleton (P) fraction. This association was accompanied by a displacement of some of the endogenous protein into the cytoplasmic (SN) fraction (Fig 3G, arrows 2).

Quantification of fractionation data supported the qualitative analysis (Fig 3H). Summing the signals of Ty1-TbMORN1 and TbMORN1 present in the cytoskeletal (P) fraction in overexpressing cells indicated that the total amount of cytoskeleton-associated protein was approximately the same as in the controls (Fig 3H, grey bar). This suggested that there are a finite number of Ty1-TbMORN1 molecules that can associate with the cytoskeleton. As the total amount of endogenous and ectopic TbMORN1 associated with the cytoskeleton is roughly the same in both overexpressing cells and controls, this suggested also that the dominant negative cellular effects are primarily due to the endogenous and ectopic TbMORN1 in the cytoplasmic (SN) fraction.

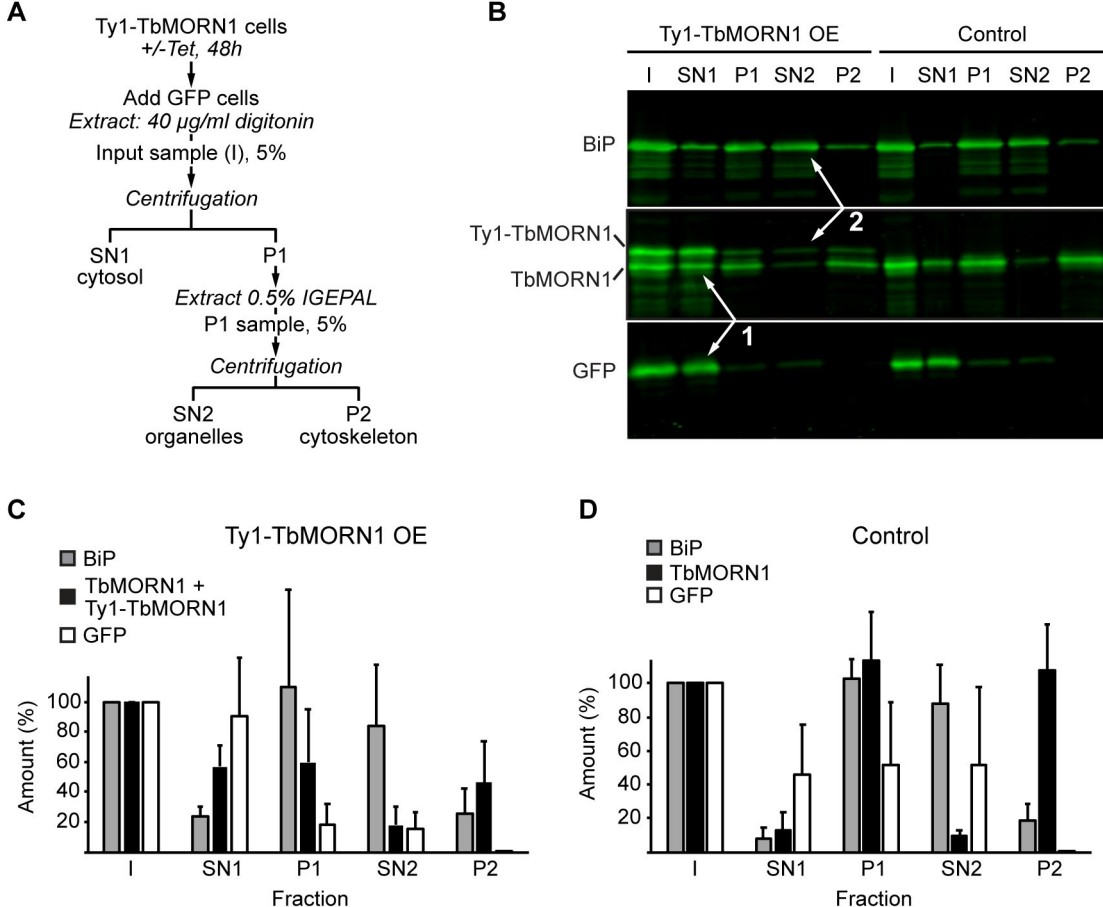

**Fig 4. Overexpressed Ty1-TbMORN1 is predominantly cytosolic. (A)** Schematic of the two-step fractionation scheme. **(B)** Immunoblots of fractions taken from control and Ty1-TbMORN1 overexpressing cells, using anti-BiP, anti-TbMORN1 and anti-GFP antibodies. Note that the membrane was cut into three strips for the immunoblot. Equal fractions (5%) were loaded in each lane. The overexpressed Ty1-TbMORN1 was predominantly extracted by digitonin and partitioned with the cytosolic GFP into the SN1 fraction (arrows 1). Very little of the remainder was subsequently extracted with non-ionic detergent into the SN2 fraction (arrows 2), with most partitioning into the cytoskeleton-associated P2 fraction. Three independent experiments were carried out using cells from three clones pooled together; an exemplary blot is shown. **(C,D)** Quantification of the immunoblots of the two-step fractionation. Data were compiled from three independent experiments, each using cells pooled from three separate clones. Bars show mean values + SD.

To determine if the cytoplasmic fraction of endogenous and ectopic TbMORN1 in the overexpressing cells was organelle-associated or cytosolic, two-step fractionations were carried out (Fig 4A). These assays involved a first extraction with digitonin to generate a cytosolic fraction (SN1), then a second extraction with IGEPAL to generate an organelle-associated fraction (SN2). Digitonin has an affinity for cholesterol and other lipids enriched in the plasma membrane, so at the right concentration it should enable the extraction of cytosol while leaving organelle membranes relatively intact [80].

The conditions for digitonin extraction were optimised using a cell line expressing cytosolic GFP as a marker (S8 Fig) [53]. A two-step extraction using first digitonin and then IGEPAL was carried out on Ty1-TbMORN1 overexpressing cells, with the cells expressing cytosolic GFP spiked in alongside them (Fig 4A). The digitonin supernatant (SN1) was enriched for cytosol, as shown by the presence of the GFP marker. The organelle-associated fraction, tracked by immunoblotting for the ER lumenal chaperone BiP, was present in the pellet (P1).

The P1 fraction was subsequently extracted using IGEPAL and the BiP marker was observed to predominantly partition into SN2 (Fig 4B). Analysing equal fractions by immunoblotting showed that the Ty1-TbMORN1 and TbMORN1 proteins not associated with the cytoskeleton were predominantly cytosolic and partitioned with the GFP marker into the SN1 fraction (Fig 4B, arrows 1).

The Ty1-TbMORN1 and TbMORN1 in the P1 fraction (Fig 4A) were not strongly extracted by IGEPAL and therefore barely present in the second supernatant (SN2), while the ER marker BiP was (Fig 4B, arrows 2). Almost all the Ty1-TbMORN1 and TbMORN1 present in P1 partitioned into the second pellet, P2. Quantification of multiple independent experiments using the three separate clones produced results consistent with the exemplary blot shown (Fig 4C and 4D). The presence of the overexpressed Ty1-TbMORN1 and displaced endogenous TbMORN1 in the digitonin supernatant (SN1) indicated that they are predominantly cytosolic.

In conclusion, the extensive studies conducted on TbMORN1 here provide no evidence that its MORN repeats are able to associate with phospholipid membranes in vivo or directly interact with phospholipid vesicles in vitro.

## High-resolution crystal structures of three MORN repeat proteins

If MORN repeats do not bind membranes, then this raises the question of what they really do and whether this other function might unify the various observations made about MORN repeat proteins to date. ALS2 has been suggested to use its MORN repeats to form an antiparallel dimer [81] and the evidence obtained here showed that TbMORN1 molecules also formed tail-to-tail dimers via their C-termini (Figs 1B and S1D). Mammalian PI(4)P 5-kinases are also dimers, implying that the MORN repeats found at the N-termini of the family of plant PIPKs might function to mediate homotypic interactions [82]. To investigate the molecular basis of TbMORN1 dimerisation, high-resolution structural studies were used.

Crystallisation trials were initially performed with TbMORN1(2–15) and TbMORN1(7–15). Diffraction data were obtained for TbMORN1(7–15) in two different crystal forms (P2$_1$ and C2) but attempts to solve the phase problem using experimental phasing approaches (multiple isomorphous replacement, multiple anomalous scattering exploiting selenium and sulphur atom signals) and molecular replacement failed due to low reproducibility of crystals, anisotropy of diffraction data, and absence of sufficient homology of TbMORN1 to other MORN repeat-containing proteins of known structure. At that point, the only MORN repeat-containing protein for which the crystal structure was solved was SETD7, a histone methyltransferase. SETD7 is predicted by the TRUST algorithm to contain up to 6 MORN repeats at its N-terminus [83]. However, the repeats display low sequence homology to both the MORN repeats of TbMORN1 and the consensus MORN repeat sequence obtained by Pfam [2]. This prevented successful use of molecular replacement as an approach.

As a new tactic, the MORN1 proteins from the apicomplexan parasites *Plasmodium falciparum* (PfMORN1) and *Toxoplasma gondii* (TgMORN1) were targeted for structure determination. Despite their evolutionary distance, they share high (57% and 54%, respectively) sequence identity with TbMORN1 (S9A and S9B Fig). CD measurements of purified recombinant protein indicated that TgMORN1, TgMORN1(7–15), PfMORN1, PfMORN1(2–15), and PfMORN1(7–15) all had an overall β-structure (S9C and S9D Fig). This agrees with bioinformatic predictions and is consistent with the data obtained for TbMORN1 (S1E Fig).

Diffracting crystals of selenomethionine-labelled PfMORN1(7–15) were obtained and its crystal structure was determined to 2.14 Å resolution using the single-wavelength anomalous dispersion (SAD) method. The structures of TgMORN1(7–15) and both P2$_1$ and C2 crystal

forms of TbMORN1(7–15) were subsequently determined to 2.90, 2.35 and 2.53 Å resolution, respectively, with the PfMORN1(7–15) structure used as a search model for the molecular replacement (Fig 5A–5C). The structures of PfMORN1(7–15), TgMORN1(7–15), and both P2$_1$ and C2 forms of TbMORN1(7–15) were refined to an R$_{work}$/R$_{free}$ of 23.0%/26.4%, 28.3%/32.2%, 23.2%/25.6% and 22.5%/28.2%, respectively (Table 1). TbMORN1(7–15) P2$_1$ (Fig 5A), C2 (S10A Fig) and TgMORN1(7–15) (Fig 5B) crystallised with one dimer in the asymmetric unit, while PfMORN1(7–15) (Fig 5C) crystallised with one subunit in the asymmetric unit and the functional dimer was formed via crystallographic symmetry axis.

All MORN1(7–15) crystal structures show subunit interactions via the C-terminal regions to form antiparallel tail-to-tail dimers, with variable inter-subunit angles and dimerisation interfaces (Figs 5A–5C and S10A). The two subunits in the TbMORN1(7–15) C2 crystal form and the TgMORN1(7–15) dimer make a rather straight assembly, while in the P2$_1$ form they display a bend of about 30˚ (S10A Fig), leading to a rod-shaped particle appearance. This dimer architecture is consistent with the limited proteolysis data, which had indicated that the N-terminal regions of the molecule are more susceptible to limited proteolysis (S1A and S1B Fig). Interestingly, PfMORN1(7–15) displays a V-shaped dimer with an inter-subunit angle of about 45˚, and incorporates a structural Zn$^{2+}$ ion at the dimer interface (Fig 5C). One crystal form of TgMORN1(7–15) also adopts the same V-shape seen for PfMORN1(7–15) (S12B Fig). Here too, a Zn$^{2+}$ ion is found at the dimer interface.

Superimposing the TbMORN1(7–15) P2$_1$ subunit with the structures of the TgMORN1(7–15) and PfMORN1(7–15) subunits over 202 C$_\alpha$ atoms yielded rmsd values of 1.0 and 1.1 Å, respectively, revealing high structural similarity between the three proteins. The common structural feature of all three subunits is an elongated and twisted β-sheet. The curved shape of each constituent MORN repeat forms a groove laterally delimited by a rim. An individual MORN1(7–15) subunit is approximately 80 Å in length and displays a longitudinal groove of about 16 Å in depth (Fig 5A).

Comparison of a single TbMORN1(7–15) subunit with known three-dimensional structures was carried out using the DALI server [84, 85]. This revealed closest similarity with the G-box domain at the C-terminus of the human CPAP protein. CPAP is a centriolar protein essential for microtubule recruitment. The G-box comprises a single elongated β-sheet with all residues being solvent-exposed and is capable of forming supramolecular assemblies [PDB entry 4LZF, [86]]. Despite the low sequence identity (10%) between TbMORN1(7–15) and CPAP, the Z-score calculated by DALI suggested a significant similarity between the two structures (rmsd over 155 superimposed C$_\alpha$ atoms = 3.9 Å, Z-score = 12).

## A structure-based redefinition of the MORN repeat

The crystal structures confirmed that each 23-amino acid MORN repeat is composed of a β-hairpin followed by a 6-residue loop that connects to the next MORN repeat. Each β-hairpin is composed of two 6-residue β-strands connected by a 5-residue loop. The MORN repeats from TbMORN1(7–15) could be readily superimposed, showing a high level of structural conservation (Fig 5D). Based on this high level of conservation, a revised sequence alignment of the TbMORN1 repeats was constructed that better reflects the structural architecture of the protein (Fig 5E, compare with Fig 1A). The alignment of repeats 7–15 was obtained directly from structural superpositions and used to bootstrap the alignment of the repeats 1–6.

The new consensus MORN repeat sequence displays three highly-conserved features: a GxG motif at the start, a conserved glycine (G) at position 10, and a YEGEW motif at positions 13–17 (Fig 5E). A slightly less conserved LxY motif is at positions 5–7. The GxG motif is at the beginning of the first β-strand, while the YEGEW motif comprises most of the second β-strand.

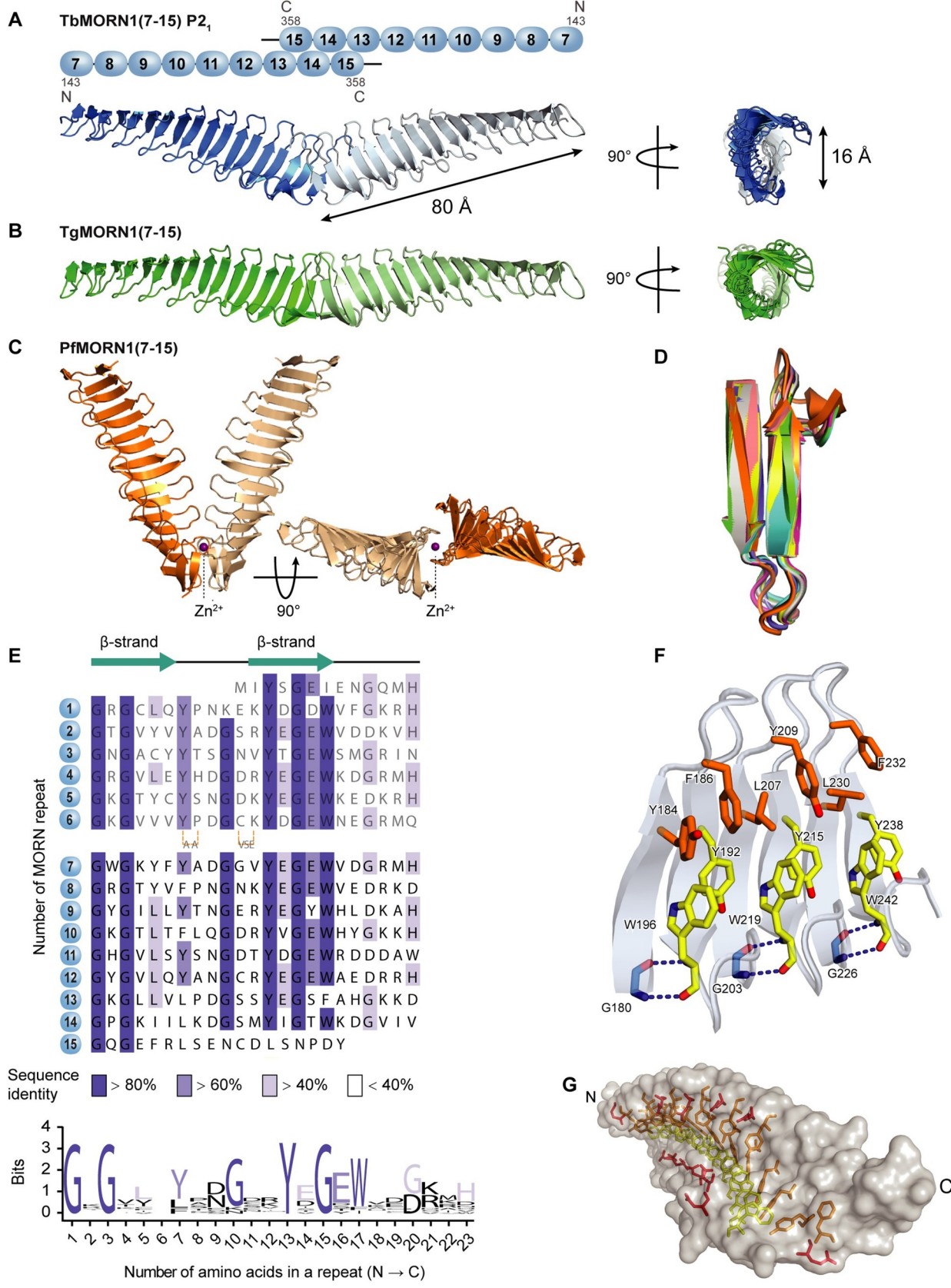

**Fig 5. High-resolution structures of MORN repeat proteins and a structural redefinition of the MORN repeat. (A)** Schematic depiction and crystal structure of the TbMORN1(7–15) dimer in its P2₁ crystal form. Amino acid numbers and N- and C-termini are indicated in the schematic. The crystal structure is shown in two orientations, with main dimensions indicated in Å. The structure contains 2 x 9 MORN repeats and is an antiparallel homodimer with the subunits arranged in a splayed tail-to-tail configuration. The secondary structure consists of exclusively antiparallel beta-strands and peripherally positioned loops, which together form a longitudinal groove through the middle of the protein. **(B)** Crystal structure of TgMORN1(7–15) shown in two orientations. The number of MORN repeats and the configuration is the same as for TbMORN1(7–15) in panel A. **(C)** Crystal structure of PfMORN1(7–15) shown in two orientations. The bound Zn²⁺ ion is labelled. The structure contains 2 x 9 MORN repeats, again in tail-to-tail configuration but with an overall V-shaped arrangement. **(D)** Alignment of all 9 TbMORN1(7–15) MORN repeats in the crystal structure reveals a high level of structural conservation. **(E)** A revised consensus MORN repeat sequence, based on the crystal structures. A new alignment of the MORN repeats in TbMORN1 is shown. Repeats 7–15 are present in the crystal structure; repeats 1–6 are inferred. Conservation of sequence identity is indicated by colour intensity. Each MORN repeat consists of a β-hairpin, built up of two 6-residue β-strands connected by a 5-residue loop. The β-hairpin is followed by a 6-residue loop that connects to the next MORN repeat. The new 23-residues long consensus MORN repeat starts with the GxG motif. **(F)** The tertiary structure of individual MORN repeats is stabilised by hydrogen bonds between the first G of the GxG motif and the W from the YEGEW motif. MORN repeat arrays are further stabilised by aromatic stacking between the highly conserved aromatic residues in the YEGEW and LxY motifs, and by T-shaped π-stacking interactions of the highly conserved Y of the YEGEW motif, which is sandwiched between the W residue of its own motif, and the W residue in the next YEGEW motif. **(G)** A single TbMORN1(7–15) subunit viewed at an oblique angle. The residues of the YEGEW and LxY motifs involved in aromatic stacking line the surface of the longitudinal groove running through the middle of the protein.

The glycine residues at position 10 are in a β-hairpin of type I, where the most commonly-observed residue at this position is a glycine [87]. The GxG motif is strictly conserved because the first G residue adopts a main chain conformation mapping to the lower right corner of the Ramachandran plot, which is exclusively allowed for glycines. The conformation of this glycine is stabilised via a main-chain hydrogen bond with the tryptophan (W) residue of the YEGEW motif (Fig 5F). The high conservation of the second glycine residue in the GxG motif is to accommodate the highly conserved neighbouring aromatic residues from the YEGEW and LxY motifs, as any larger side chain would create steric clashes. The tyrosine and tryptophan side chains of the YEGEW and LxY motifs provide a textbook example of aromatic stacking, filling up the groove and stabilising the tertiary structure of the TbMORN1 subunit (Fig 5F and 5G). The highly conserved tyrosine of the YEGEW motif is sandwiched in a T-shaped π-stacking interaction between two tryptophan residues, one from its own motif and the second from the following YEGEW motif (Fig 5F). The tyrosine of the LxY motif is stabilised via hydrophobic and/or aromatic stacking interactions with the leucine residues in its own and the subsequent LxY motif.

Three MORN repeats of SETD7 display similar general features, but differ in details, and can be aligned with TbMORN1 repeat 7 over 22–23 aligned Cα atoms with an rmsd of 2.3, 1.5, and 1.9 Å respectively (S10B Fig). The first glycine residue in the SETD7 MORN repeats is conserved with that in TbMORN1, while tyrosine, phenylalanine, and valine replace the tryptophan of the YEGEW motif (S10C Fig).

## TbMORN1 forms an extended antiparallel dimer with overall negative charge

The TbMORN1(7–15) antiparallel dimer displays a two-fold symmetry perpendicular to the longitudinal axis of the assembly (Fig 5A). Analysis of amino acid conservation derived from a sequence alignment of representative MORN repeat-containing proteins showed a well conserved stretch of residues in the groove (S11A Fig). Due to the twofold symmetric nature of the quaternary structure assembly, structural and surface properties are displayed on opposite sides of the elongated dimeric particle and perpetuated symmetrically along the rims (S11 Fig).

The TbMORN1 dimer displays an overall negative electrostatic potential with a central positively-charged pocket (S11B Fig). The overall negative charge of TbMORN1(7–15) and the lack of pronounced positive patches that could serve as binding sites for negatively-charged

phospholipid polar heads is consistent with the negative binding data for lipid membranes (S11B and S11C Fig).

TbMORN1(7–15) formed an extended antiparallel dimer in both crystal lattices, while TgMORN1(7–15) adopted this architecture in one crystal form. The extended dimer interface is built from residues in MORN repeats 12–15, which connect the two subunits in an antiparallel, tail-to-tail orientation (Fig 5A–5C). In both crystal forms of TbMORN1(7–15), the central core of the dimer interface is broadly similar (Figs 6A and S12A). Due to the twofold symmetry, the majority of the stabilising interactions are duplicated and build up an extended dimer interface.

The tightest overlap between the two subunits occurs through MORN repeats 14 and 15. A series of hydrophobic and aromatic π-stacking interactions between residues from repeats 14 and 15, together with hydrogen bonds across the edges, stabilise the dimer (Figs 6A and S12A). Furthermore, the very negatively-charged C-terminal region of one TbMORN1 subunit forms an arch above the positively-charged platform contributed by Lys316 and Arg293 (S11B

**Fig 6. Dimerisation interfaces of TbMORN1(7–15) and PfMORN1(7–15). (A)** The dimerisation interface of TbMORN1(7–15) P2₁ involves residues from MORN repeats 12–15, which stabilise the dimer interface via aromatic π-stacking (Tyr330 and Phe345 from the respective subunits), hydrophobic interactions (Leu301, Leu347, Ile339, Ile322, Leu324), and additionally via hydrogen bonding interactions at the edges of the dimer interface. In comparison to the TbMORN1(7–15) C2 crystal structure, there are no disulphide bridges stabilising the dimerisation interface of TbMORN1(7–15) P2₁. **(B)** The dimerisation interface of the V-shaped PfMORN1(7–15) dimer is smaller and is additionally stabilised by the incorporation of a structural $Zn^{2+}$ ion, which is tetrahedrally coordinated by Cys306 and Asp309 residues from each respective subunit. Thr311 holds the side chain of Asp309 in the appropriate orientation. The dimer interface is additionally stabilised by symmetric hydrogen bonding between the Thr311 pair, aromatic stacking between the Phe330 pair, a hydrophobic cluster formed by Leu328, Val331 and Leu336, two salt bridges between Lys322 and Glu308 from the respective subunits, anion-π interaction of a side chain of Glu308 with Phe304, and a combination of aromatic stacking (His334 pair) and polar interactions (His334, Asn332) at the vertex of the dimer. **(C)** An electrostatic map calculated for a single sub-unit of TbMORN1(7–15). Vacuum electrostatics were calculated in PyMOL, where red denotes -65.7 kT and blue denotes +65.7 kT. The structure on the right shows the predicted effect of two point mutations, R293A from MORN repeat 13, and K316A from MORN repeat 14. The mutations are expected to result in the loss of a positively-charged patch close to the dimer interface, and consequently disrupt the dimerisation of the TbMORN1(2–15) constructs.

Fig). Two residues from MORN repeat 14—Lys325 (subunit A) and Asp326 (subunit B)—form a salt bridge, which further stabilises the dimer interface.

In the P2$_1$ and C2 TbMORN1(7–15) crystal forms, the dimer interfaces occupy surface areas of 747 Å$^2$ and 966 Å$^2$, respectively. Calculations of gain in solvation free energy ($\Delta^i$G) upon dimer formation for TbMORN1(7–15) C2 and P2$_1$ performed with the PDBePISA package [88] yielded values of -20.1 kcal/mol and -11.3 kcal/mol, respectively, with corresponding p-values of interface specificity 0.01 and 0.08. $\Delta^i$G values lower than -10 to -15 kcal/mol and p-values lower than 0.5 are significant for stable protein dimers, indicating highly specific dimerisation interfaces [88].

In the C2 crystal form, the dimer is additionally stabilised by two disulphide bridges formed between Cys351 at the C-terminus of repeat 15 and Cys282 from the β-hairpin loop of repeat 12 (S12A Fig). In the P2$_1$ crystal structure, the position of the loop differs from that in the C2 dimer and keeps the C$_\alpha$ atoms of Cys351 and Cys282 at a distance of 11.7 Å, preventing disulphide bond formation. Here, the side chain of Cys351 is engaged in a polar interaction with Asp303 (Figs 6A and S12A).

In comparison to the TbMORN1(7–15) C2 crystal form, the extended TgMORN1(7–15) structure has an approximately 1.5-times smaller dimer interface (601 Å$^2$), which is contributed by residues from MORN repeats 13–15 only (Figs 5B and S12C). This is closer in size to the 747 Å$^2$ interface of the P2$_1$ crystal form of TbMORN1(7–15). The final C-terminal part of the protein seems to be flexible and does not engage in the stabilisation of the dimer. In comparison to TbMORN1(7–15), the dimer interface of TgMORN1(7–15) is not built around aromatic stacking but instead employs hydrophobic interactions between Phe350 and neighbouring small hydrophobic residues, such as Leu335 and Leu327 (S12C Fig).

A retroactive validation of the dimer interface came from the earlier PI(4,5)P$_2$-binding work. Mutagenesis of repeat 14 had produced a mixture of monomers and dimers, while the simultaneous mutagenesis of two candidate sites in repeats 13 and 14 had resulted in a pure monomer population (S4B Fig). Analysis of the interaction and electrostatic maps of TbMORN1(7–15) and its mutagenised variants clearly showed that Arg293 and Lys316, residing in repeats 13 and 14 respectively, are crucial for maintaining TbMORN1 in a stable dimeric state through electrostatic interactions (Fig 6C). Lys315 (in MORN repeat 14) does not appear to be directly involved in the dimerisation interface but could peripherally contribute to the stabilisation of the C-terminal region of TbMORN1 through its electrostatic potential. Residues Arg292, Lys296 (both in MORN repeat 13), and Arg321, Lys325 (both in MORN repeat 14) were mapped to the outer surface of the dimer and therefore do not take part in the dimerisation interface (Figs 6A and S12A).

The transition of TbMORN1(2–15)$^{Mut14}$ from a dimeric to a mixed monomer/dimer population is therefore a direct consequence of the single point mutation K316A, whereas the complete abrogation of dimerisation observed for the TbMORN1(2–15)$^{Mut13+14}$ double mutant can be attributed to a synergistic effect of both R293A and K316A point mutations (Figs 6C and S4B), thereby verifying their contribution to dimer formation.

## The V-shaped and extended dimer forms of apicomplexan MORN1 proteins

Unlike the extended dimers of TbMORN1(7–15) and TgMORN1(7–15), the V-shaped PfMORN1(7–15) is mainly stabilised by a single Zn$^{2+}$ ion incorporated into the core of its dimer interface, spanning over 665 Å$^2$ (Figs 5C and 6B). The Cys306 and Asp309 residues from each subunit tetrahedrally coordinate the Zn$^{2+}$ at expected distances (2.32 Å for Zn$^{2+}$-S and 1.94 Å for Zn$^{2+}$-O). In addition, residues from repeats 13–15 are involved in stabilising

the dimer via a combination of hydrophobic, polar, and electrostatic interactions across the subunits.

Although TgMORN1(7–15) was predominantly found as an extended dimer (Fig 5B), a V-shaped form similar to that of PfMORN1(7–15) was also observed in the crystal lattice (S12B Fig). The two V-shaped dimers share the same coordination sphere of a $Zn^{2+}$ ion, which in TgMORN1(7–15) is provided by the Cys305 and Asp308 residues (S12D Fig). The latter residue in turn interacts with the Ser310 from the other subunit. While the salt bridge and anion π-interactions are also conserved between the two V-shaped dimers, the aromatic stacking present in the core of the PfMORN1(7–15) dimer interface is functionally replaced in the TgMORN1(7–15) V-shaped dimer by a series of aromatic stacking interactions at its vertex.

Although Asp residues 303 (TbMORN1), 308 (TgMORN1), and 309 (PfMORN1) are conserved in all the three proteins, the coordination of a $Zn^{2+}$ ion clearly demands the presence of both cysteine and aspartate residues. Such pairs are present in PfMORN1 and TgMORN1, but not in TbMORN1, where the cysteine residue at the corresponding position is replaced by Leu301. The coordination residues map to the β-hairpin of MORN repeat 13, which in TgMORN1 and PfMORN1, but not TbMORN1, contains an insertion of a glutamate residue —Glu307 and Glu308, respectively. Taking part in an anion π-interaction with a phenylalanine residue (Phe303 and Phe304, respectively), the resulting Glu-Phe pairs effectively stabilise the TgMORN1 and PfMORN1 dimers in their V-shaped form. Moreover, these very same glutamate residues further stabilise the two V-shaped dimers by being involved in a salt bridge with lysine residues (Lys321 in TgMORN1 and Lys322 in PfMORN1). At the equivalent position in TbMORN1, Lys316 does not participate in a salt bridge but rather points towards the negative patch at the C-terminal part of the other subunit and stabilises the extended dimer via electrostatic interactions.

To see whether it was possible to predict if a MORN repeat protein forms either extended or V-shaped dimers or both, a comparative sequence analysis was carried out. The sequences of 15 selected MORN repeat proteins from various protist lineages were aligned with the C-terminal parts of TbMORN1, TgMORN1, and PfMORN1 encompassing repeats 12–15 (S13 Fig). The residues in the crystal structures of TgMORN1(7–15) and PfMORN1(7–15) forming the $Zn^{2+}$-coordination sphere and anion π-interaction were taken as a fingerprint for a V-shaped dimer. All sequences of MORN repeat proteins from kinetoplastids contain a leucine residue (Leu301 in the case of *T. brucei*) at the position of the coordinating Cys residue. They additionally lack the Phe-Glu pair that flank the coordinating Cys residue and form the anion π-interaction. As these residues are essential for V-shape dimerisation, the kinetoplastid proteins should form extended dimers only. Conversely, all protein sequences in the dataset from apicomplexans (*Toxoplasma gondii*, *Plasmodium falciparum*, *Gregarina niphandrode*s, *Babesia microti*, *Cryptosporidium parvum*, *Eimeria acervuline*, *Theileria equi*) contained these cysteine, phenylalanine, and glutamate residues. This suggests that all these apicomplexan proteins can adopt both extended and V-shaped forms. In addition, the sequences from the alveolates *Symbiodinium microadriaticum* and *Perkinsus marinus*, and the stramenopile *Aureococcus anophagefferens* also contain these three residues, implying that they too should adopt both extended and V-shaped conformations. Apicomplexans belong to the Alveolata clade, and both alveolates and stramenopiles are in the SAR supergroup [89]. This suggests that the ability to adopt two conformations might have arisen within this specific clade and possibly explains its absence from the kinetoplastid sequences, as kinetoplastids are excavates. The functional implications of this built-in structural plasticity seemingly modulated by the presence of zinc ions remain intriguing.

## MORN1 proteins adopt extended conformations in solution

The crystal structures were all consistent with the earlier results of the cross-linking mass spectrometry experiments carried out on TbMORN1 in solution, which had indicated close proximity between repeats 13–15 (S1 Table). To investigate the presence of V-shaped and extended dimers in solution, all three MORN1 proteins were structurally analysed using small-angle X-ray scattering (SAXS). SAXS analysis of TbMORN1(7–15) and TgMORN1(7–15) indicated an extended dimer in solution and the crystal structures could be docked into the calculated molecular envelopes without difficulty (Figs 7A, 7B and S12E). The SAXS analysis of PfMORN1(7–15) indicated an extended structure in solution however, similar to that seen for TbMORN1(7–15) and TgMORN1(7–15) (Figs 7C and S12E) and different from the V-shaped form observed in the crystal lattice (Fig 5C). This supports the prediction that TgMORN1 and PfMORN1 are capable of adopting two different conformations. Subsequent SAXS experiments produced a molecular envelope for TbMORN1(2–15) into which an extrapolated version of the model of TbMORN1(7–15) could be docked as an extended dimer (Figs 7D and S12E).

Rotary shadowing EM on TbMORN1(2–15) produced results consistent with the SAXS analysis, showing small kinked rods approximately 25 nm in length (Fig 7E). The population was homogeneous, consistent with the monodispersity of this sample observed by SEC-MALS (Fig 1B). Comparison of the maximal dimer dimension ($D_{max}$) for TbMORN1(2–15) obtained from SAXS and EM showed a very good agreement between the two values of 250–260 Å. Rotary shadowing EM was also carried out on the small amount of TbMORN1(1–15) that eluted from the SEC column (Fig 7F and 7G). TbMORN1(1–15) was far more heterogeneous than TbMORN1(2–15), consistent with the polydispersity observed by SEC-MALS (Fig 1B). In addition to single kinked rods, longer filamentous assemblies of varying length were occasionally observed (Fig 7F). Rarely, much larger meshlike assemblies of full-length TbMORN1 could be observed (Fig 7G), offering a tantalising clue into the higher-order assembly properties of the protein. These properties will be investigated in future work.

## Discussion

MORN repeats were first named almost 20 years ago in a paper identifying the junctophilin protein family [1]. While there is abundant evidence that junctophilins associate with the plasma membrane and that the MORN repeat-containing region is likely to mediate this, there is to the authors' knowledge no report demonstrating that the junctophilin MORN repeats directly interact with lipids [1, 12, 13, 90–93].

This evidence that the MORN repeat-containing region of junctophilins mediates plasma membrane targeting has been repeatedly cited as showing that MORN repeats directly bind lipids. Warnings that the function of MORN repeats has not really been elucidated and that extant data are frequently contradictory, have been largely overlooked [17]. We therefore set out to test the lipid-binding hypothesis by using a protein composed exclusively of MORN repeats, TbMORN1.

Early experiments utilising isolated lipids in PIP strips and fluorescence anisotropy assays suggested an ability of TbMORN1 to bind specific phospholipids. These results could never be confirmed in liposome pelleting assays however and the quantitative molecular biophysics data strongly suggest that the observed binding was to the aliphatic chains of the isolated lipids.

Given that TbMORN1 is a cytoskeleton-associated protein, it is hard to imagine how it would be able to access the hydrophobic chains of membrane-embedded phospholipids in vivo, although it is localised directly under the cytoplasmic leaflet of the plasma membrane.

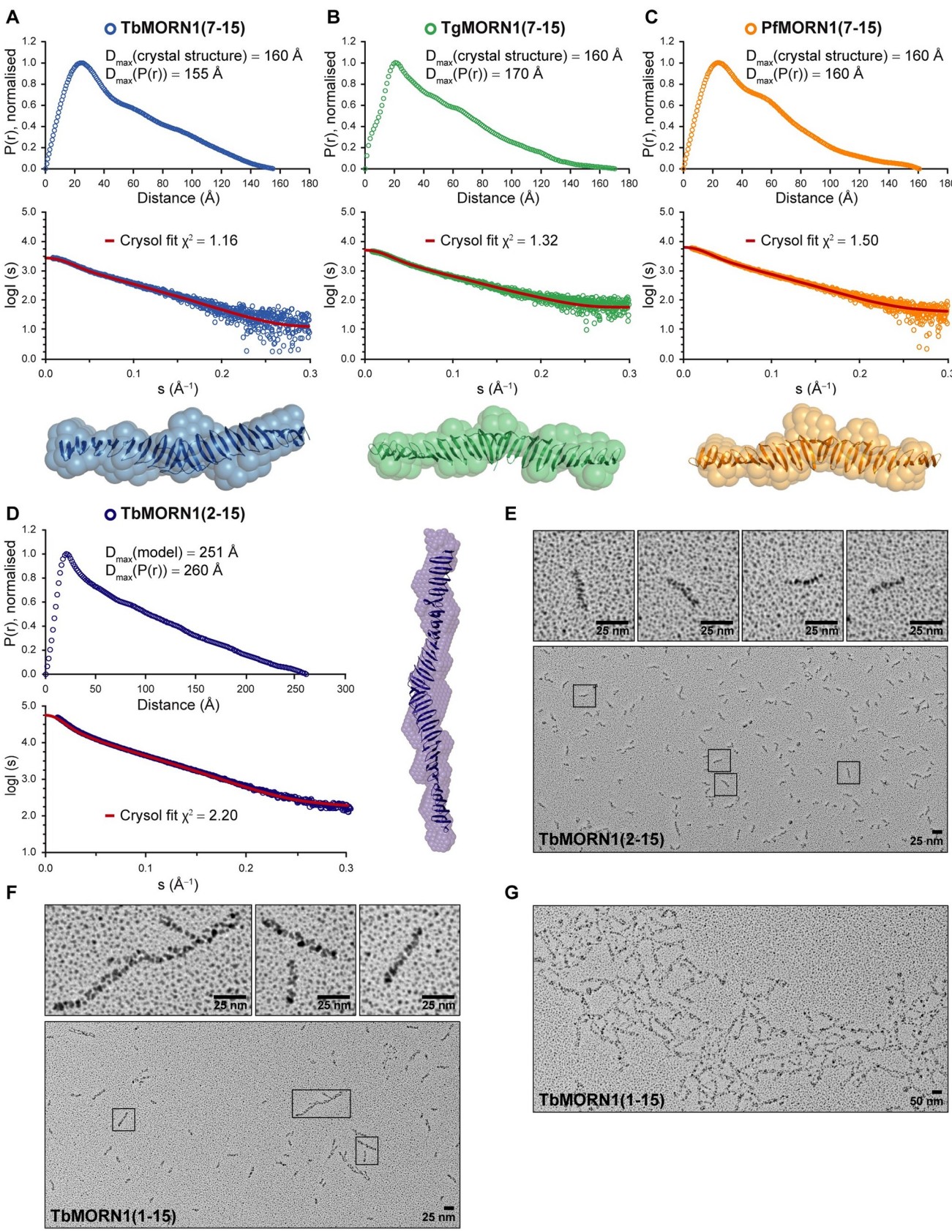

**Fig 7. MORN1 proteins form extended dimers in solution as assessed by SAXS and EM experiments. (A-D)** SAXS experiments on: **(A)** TbMORN1(7–15), **(B)** TgMORN1(7–15), **(C)** PfMORN1(7–15), and **(D)** TbMORN1(2–15). For each respective protein, the results include a P(r) plot with derived experimental $D_{max}$ value compared with a $D_{max}$ value derived from the structure, an experimental SAXS scattering data with a fit calculated by the Crysol programme, and a SAXS-based ab initio molecular envelope. In the case of TbMORN1(2–15), the theoretical $D_{max}$ value was derived from a structural model, which was generated by spiking the TbMORN1(7–15) structure with additional structures of individual TbMORN1(7–15) subunits. **(E-G)** EM with rotary shadowing of TbMORN1(2–15) and full-length TbMORN1. **(E)** TbMORN1(2–15) forms a homogenous population of extended dimers of approximately 25 nm in length (see insets for individual examples). **(F)** Full-length TbMORN1 is heterogeneous and includes rare filaments of 175–200 nm in length (first inset) and individual dimers (second and third inset). **(G)** Higher-order assemblies of full-length TbMORN1 assembled in a mesh-like structure. Magnification, 71,000x; scale bars, 25 nm, 50 nm as indicated. n (independent replicates) = 2, n (biological replicates) = 2.

The electrostatic profile of TbMORN1(7–15) is also not suggestive of membrane binding, with a strong overall negative charge profile (S11B Fig).

Unexpectedly, purified recombinant TbMORN1(1–15) was found to co-purify with PE. Bacteria expressing TbMORN1(1–15) showed elevated levels of PE, which might suggest that TbMORN1 was binding and sequestering the lipid, requiring them to upregulate synthesis (Fig 2A and 2C). Almost no PE was found to co-purify with TbMORN1(10–15) and none was detected with the apicomplexan MORN1 proteins. This suggests that the first 9 MORN repeats of TbMORN1 might be a major interaction area for PE.

No co-purification of PE was seen with the apicomplexan MORN1 proteins despite >50% identity at an amino acid level. This lipid-binding activity therefore seems specific to TbMORN1. The elevation of PE levels seen in bacteria expressing TbMORN1(1–15) is highly unusual and the authors do not currently have a good explanation apart from invoking lipid sequestration and transport to specific sites [94, 95]. If the PE co-purifying with recombinant TbMORN1(1–15) was just carry-over, then the lipid profile should resemble that of total bacterial cellular lipids, which is not the case. Furthermore, the bound PE moieties display a narrow range of aliphatic chains, suggesting specificity of binding/recognition.

Lipids in trypanosomes have, in general, much longer aliphatic chains than those in bacteria. In *T. brucei*, PE accounts for around 10–20% of total lipid, with aliphatic chains of 36:0 being the dominant isoform [47]. In *E. coli*, PE is the predominant zwitterionic lipid and accounts for around 80% of total lipid [96]. As the length of the side chains in bacterial lipids is predominantly 14 and 16 carbon atoms [97], PE molecules with 16 carbon atoms chains are the most common. Notably, TbMORN1(1–15) overexpressed in *E. coli* co-purified with PE species of much greater length (30:0–36:2) (Fig 2A), which was also reflected in increased production of these PE species in overexpressing bacteria (Fig 2C). Taken together, these data strongly suggest the specificity of TbMORN1(1–15) towards PE species with aliphatic chains of a length characteristic for *T. brucei*.

Neither PE nor any other lipids were observed in the crystal structure of TbMORN1(7–15). This might be due to the fact that TbMORN1(7–15) does not contain the N-terminal part of the molecule. Alternatively, the lysine methylation step used to enhance crystallisation might have prevented electrostatic interactions with lipid molecules due to the loss of positive charge.

Due to the twofold symmetry of both extended and V-shaped MORN1 dimers, the dimeric particle displays the conserved groove on opposite sides, therefore excluding this region for interaction with the membranes and leaving as an option the rims lining the groove. In the extended dimers, the two rims display a concave and a convex curvature (S11 Fig), suggestive of membrane sculpting BAR domain proteins which interact with membranes mainly via non-specific electrostatic interactions [98, 99]. These curved surfaces in MORN1 dimers nevertheless do not display a pronounced positive charge, nor the typical membrane insertion motifs characteristic for BAR domain proteins, thus refuting the hypothesis for membrane binding. In conclusion, the quaternary structure architecture together with surface properties of the MORN1 dimers do not support membrane binding.

It is important to note that phospholipid binding and membrane binding are two separate things. Not interacting with membranes might not preclude the TbMORN1 protein extracting PE or other phospholipids out of membranes without stably interacting with the membranes itself. A similar hypothesis would be that TbMORN1 functions as a carrier/transporter of lipids with aliphatic chains within a certain length, as for example found for non-vesicular phosphatidylserine transport by oxysterol-binding proteins from the site of synthesis to the site of biological activity [100]. What would be the corresponding physiological role in the context of the flagellar pocket in trypanosomes is difficult to envisage at present. For now, the simplest interpretation remains that there is no interaction of TbMORN1 with lipid membranes and consequently no physiological interaction with membrane-embedded phospholipids either. These data have clear implications for other MORN repeat proteins. Junctophilins do appear to bind lipids but it is not clear if their MORN repeats are mediating this or are merely within the region/domain involved [7, 101].

Based on the data presented in this study, the presence of MORN repeats in a protein should not be taken as indicative of lipid binding or lipid membrane binding without experimental evidence. Equally, evidence of binding from PIP strips alone should be interpreted with caution.

The structural studies presented here make a case that one conserved function of MORN repeats is in homotypic interactions and possibly also in higher-order assembly. The role of MORN repeats in heterotypic interactions has recently been demonstrated at a structural level [102]. In the case of TbMORN1, previous work using proximity-dependent labelling (BioID) has already established that it has a number of direct and indirect binding partners in vivo [24].

All three MORN1 proteins analysed here formed tail-to-tail dimers via their C-termini with the subunits aligned in an antiparallel arrangement. The all-β structure of the proteins folds in a twisting elongated shape, with a groove lined with aromatic side chains running longitudinally through it. While this manuscript was in preparation, Li et al. published a high-resolution structure of the MORN4/retinophilin protein in complex with the myosin 3a tail. Binding was mediated by this central groove, showing that it can be engaged in very high-affinity protein-protein interactions [102].

The three high-resolution structures described in this work are amongst the first canonical MORN repeat proteins to be detailed. Given that the structures are from representatives of two of the five eukaryotic supergroups—the excavates, and the SAR (stramenopiles, alveolates, Rhizaria) clade—this demonstrates how the fundamental structure of the MORN repeat has been conserved over evolutionary time.

The crystal structures in extended conformations were consistent with the lower-resolution SAXS data obtained on the same proteins in solution (Figs 7A-7C and S12E, S1 Table). Based on the crystallographic data, the apicomplexan MORN1 proteins appear capable of forming both extended and V-shaped conformations. The key residues mediating the dimer interface in the V-shaped form were defined, and shown to be conserved throughout the Apicomplexa. It thus seems possible that all apicomplexan MORN1 proteins can adopt these two conformations.

MORN1 proteins in Apicomplexa have been shown to be a key component of the basal complex, which undergoes a constriction event at the end of the cell division cycle [4, 103–105]. It is therefore tempting to speculate that these extended (Zn-free) and V-shaped (Zn-bound) conformations represent the pre- and post-constriction states of the MORN1 proteins in the basal complex. If so, this would constitute a remarkable rearrangement of the subunits in the dimer, which would move through about 145˚ (S1 Video). It is interesting in this context

to note that high levels of Zn ions are required to support the growth of the blood-stage *P. fal-ciparum* parasite [106].

The high-resolution structures also enabled a new structure-based consensus sequence for a MORN repeat to be defined (Fig 5E). A single MORN repeat forms a β-hairpin, one of the conserved building blocks of structural biology, followed by a long tail. The structural basis for the conservation of individual residues within the MORN repeat has been defined (Fig 5E and 5F). These structures strongly argue that an individual MORN repeat is longer than 14 amino acids. Despite the lower level of sequence conservation in the tail following the hairpin, a 23-amino acid length seems the most parsimonious definition from a structural perspective.

One interesting consequence of this redefinition of the repeat is that it suggests that full-length TbMORN1 begins with an incomplete repeat, a feature also noted in the MORN4/reti-nophilin structure [102]. The (M)IYSEGE residues at the very N-terminus are predicted to form only a single β-strand rather than a complete hairpin (Fig 5E). Li et al. suggested that this incomplete repeat could function as a capping element, but given that the first repeat in TbMORN1 is critical for oligomerisation leading to formation of filamentous structures, an alternative explanation is possible: the N-terminus of a second TbMORN1 molecule (itself encoding a single β-strand) could complete this hairpin through intermolecular interactions. Oligomerisation would thus be driven by a "split-MORN repeat" mechanism where a complete hairpin is formed by the interaction of the N-termini of two proteins. Further work will be needed to test this hypothesis. The concluding data presented here suggest that TbMORN1 uti-lises this oligomerisation capacity to build mesh-like assemblies, which can reach considerable size in vitro (Fig 7G). These mesh-like assemblies may reflect the endogenous organisation of the protein in vivo, where a number of binding partners have already been identified [24]. It remains possible that these assemblies are capable of direct or indirect membrane association and could potentially utilise lipid-binding properties to facilitate lipid exchange. The biophysi-cal properties of these meshworks, as well as their protein-protein interactions and possible lipid-carrying roles, are likely to be fruitful avenues of investigation.

## Supporting information

**S1 Fig. Low-resolution structural analysis of TbMORN1. (A)** Full-length TbMORN1 with an N-terminal HisTag was subjected to limited proteolysis using proteinase K (red), trypsin (green), and chymotrypsin (cyan) at protease:protein ratios (w/w) as indicated on each panel. Samples were resolved by SDS-PAGE and selected bands (labelled boxes) corresponding to proteolytic products were excised and analysed by mass spectrometry. Control (Contr.) corre-sponds to protein without protease treatment. **(B)** Mass spectrometry analysis of the excised proteolytic products indicated in panel A. Peptides identified and mapped by mass spectrome-try are shown as dark grey boxes; a schematic of the full-length construct is shown above, with individual MORN repeats labelled. Note that the proteolytic products show progressive degra-dation from their N-termini, while the C-terminal part is stable. **(C)** Coomassie-stained SDS-PAGE gel showing purified recombinant TbMORN1 truncations. **(D)** SEC-MALS traces of TbMORN1(2–15), (7–15), and (10–15). Chromatographic separation was done using a Superdex 200 Increase 10/300 GL column, void volume 7.2 ml. The three proteins all eluted as dimers. Schematics are shown underneath. **(E)** Far-UV CD profiles of TbMORN1, TbMORN1 (2–15) and (10–15). A positive peak at 195 nm and a negative one at 218 nm demonstrated that the constructs are all β-proteins. The secondary structure content predictions for each construct were calculated in BeStSel and are shown below the CD graph.
(TIF)

**S2 Fig. TbMORN1 interacts with phospholipids but not liposomes. (A)** Purified recombinant TbMORN1 binds to multiple phospholipid species in protein-lipid overlay assays. PIP strips were incubated with purified recombinant TbMORN1(1–15) protein, and bound proteins were detected by immunoblotting with anti-strep antibodies. Abbreviations: PI(n)P, phosphatidylinositol (n) phosphate; PA, phosphatidic acid; LPA, lysophosphatidic acid; LPC, lysophosphatidylcholine; PI, phosphatidylinositol; PE, phosphatidylethanolamine; PC, phosphatidylcholine; S(1)P, sphingosine-1-phosphate; PS, phosphatidylserine. Data were obtained from 3 independent experiments using 2 biological replicates; an exemplary blot is shown. **(B)** PIP strip overlaid with the PH domain of PLCδ, a positive control for PI(4,5)$P_2$ binding. Data were obtained from 3 independent experiments using 3 biological replicates; an exemplary blot is shown. The PIP strips presented here were exposed to the light source for the same time. **(C)** Fluorescence anisotropy measurements of 0.1 μM BODIPY TMR-PI(4,5)$P_2$-C16 in the presence of TbMORN1(2–15), (7–15) and (10–15). All three truncations of TbMORN1 interacted with the 16-carbon PI(4,5)$P_2$ with binding affinities in the low micromolar range. Data obtained from 3 independent experiments using 3 biological replicates, with 10 technical replicates for each experiment. Traces show mean values, bars are s.e.m. **(D)** Liposome co-sedimentation assay. POPC liposomes containing 0, 5, 10, 15 and 20% of porcine brain PI(4,5)$P_2$ were incubated with 10 μM TbMORN1(2–15). TbMORN1(2–15) was found in both pellet (P) and supernatant (SN) fractions but did not increase proportionally to PI(4,5)$P_2$ concentration. The positive control, Doc2B, bound PI(4,5)$P_2$ in a concentration-dependent manner, with a shift from SN to P fractions proportional to the increase in % of PI(4,5)$P_2$ present in the liposomes. Data were obtained from 2 independent experiments using 2 biological replicates; an exemplary blot is shown. **(E)** Quantification of the liposome pelleting assays. The amount of protein in the pellet fraction is presented relative to the amount present in the pellet fraction of the no liposome condition.
(TIF)

**S3 Fig. TbMORN1 constructs interact with PI(4,5)P2 in native gel bandshift assays. (A)** Native gel electrophoresis of TbMORN1(2–15) and **(B)** TbMORN1(10–15) in the presence and absence of PI(4,5)$P_2$, PI(4)P and PI, all labelled with BODIPY TMR fluorescent dye. α-actinin served as a positive control of PI(4,5)$P_2$ binding. Data obtained from two independent experiments, each using a different biological replicate.
(TIF)

**S4 Fig. Mutagenesis of putative PI(4,5)$P_2$ binding sites in TbMORN1(2–15) has no effect on binding. (A)** Liposome co-sedimentation assay performed on TbMORN1(2–15) in the presence of POPC liposomes containing 20% of porcine brain PI(4,5)$P_2$ and 0 or 40% cholesterol. The excess cholesterol was expected to promote local high concentrations of PI(4,5)$P_2$ on the surface of the liposomes. To assay for the effect of curvature, two batches of liposomes containing 20% PI(4,5)$P_2$ and 40% of cholesterol were tested, with the diameter of the liposomes being either 100 or 400 nm. No significant co-sedimentation of TbMORN1(2–15) and PI(4,5)$P_2$-containing liposomes was observed. The positive control, 10 μM Doc2B was predominantly found in the pellet (P) fractions. **(B)** SEC-MALS profiles of TbMORN1(2–15) and its mutagenised variants. Residues comprising the putative PI(4,5)$P_2$-binding sites in MORN repeats 13 and 14 were mutated to alanines. Mutagenesis of repeat 13 (Mut13) did not result in any change to the dimeric status of the protein. However, mutagenesis of repeat 14 (Mut14) resulted in a mixture of monomers and dimers being eluted, while mutagenesis of both repeats (Mut13+14) resulted in monomeric protein. Chromatographic separation was done using a Superdex 200 Increase 10/300 GL column, void volume 7.2 ml. **(C)** Far-UV CD profiles of TbMORN1(2–15) and its putative PI(4,5)$P_2$-binding mutants. The constructs remained β-

proteins despite the site-directed mutagenesis. **(D)** Fluorescence anisotropy measurements of TbMORN1(2–15) and its putative PI(4,5)P$_2$-binding mutants, measured in the presence of 0.1 μM BODIPY TMR-PI(4,5)P$_2$. All constructs showed good interaction with the fluorophore-conjugated PI(4,5)P$_2$.
(TIF)

**S5 Fig. TbMORN1 interacts with the aliphatic chains of phosphoinositides in fluorescence anisotropy assays. (A)** Fluorescence anisotropy measurements of TbMORN1(2–15) in the presence of BODIPY TMR-labelled PI(4,5)P$_2$ C16, PI(4,5)P$_2$ C6, PI(3,4)P$_2$ C6, PI(3,5)P$_2$ C6, PI (4)P C6 and PI C6. Binding was only observed with the 16-carbon PI(4,5)P$_2$. **(B)** Fluorescence anisotropy measurements of TbMORN1(2–15) measured in the presence of BODIPY Fluorescein-labelled PI(4,5)P$_2$ C16 and PI(3,4,5)P$_3$ C16. Both 16-carbon lipids bound equally well. **(C)** Comparison of TbMORN1(2–15) binding to 16-carbon PI(4,5)P$_2$ and PI(3,4,5)P$_3$ with two positive controls, respectively Doc2B and Akt.
(TIF)

**S6 Fig. TbMORN1(2–15) does not bind to liposomes in vitro. (A-C)** Liposome pelleting assays using sucrose-loaded vesicles (SLVs). TbMORN1(2–15) was purified in the absence **(A)** or presence **(B)** of Triton X-100 in the lysis buffer. The purified proteins were incubated with SLVs, which were then pelleted by centrifugation. Supernatant (SN) and pellet (P) fractions were analysed by SDS-PAGE using Coomassie staining. The SLVs were made from commercial lipids with an excess of either PE or PI(4,5)P$_2$, and also reconstituted from purified whole-cell trypanosome lipids (Tb). A no-SLV condition was included as an additional negative control. **(C)** The PH domains of PLCγ was used as a positive control for PI(4,5)P$_2$ binding. As expected, the PLCγ PH domain co-sedimented with PI(4,5)P$_2$-containing SLVs and was entirely present in the P fraction in this condition (arrow). The recombinant TbMORN1 proteins remained in the SN fraction in all conditions. **(D-F)** Liposome microarray analysis. Microchips carrying giant unilamellar vesicles (GUVs) with lipids of interest at three different concentrations (2, 5 and 10 mol %) were incubated with purified recombinant EGFP-Tb-MORN1(2–15). No significant binding was observed to either phosphoinositide lipids or PE. **(D)** n (independent replicates) = 7, bars show standard deviation. **(E)** n (independent replicates) = 3, bars show standard deviation. **(F)** Microchip incubated with PLC-δ1 PH domain as a positive control. A specific and concentration-dependent binding between the PLC-δ1 PH domain and DOPI(4,5)P$_2$ and SM was observed. DOPC, a carrier lipid, was used as an internal negative control of binding, as well as a marker for tracking positions of liposomes on a given microarray. n (independent replicates) = 1, bars show standard deviation. Note the different scales on the y-axes of the three charts.
(TIF)

**S7 Fig. Overexpression of untagged TbMORN1 causes a dominant negative phenotype. (A)** Inducible overexpression of untagged TbMORN1. Immunoblot of whole-cell lysates from three separate clones overexpressing untagged TbMORN1 from an ectopic locus. TbMORN1 was detected using anti-TbMORN1 antibodies; PFR1,2 were used as a loading control and detected using anti-PFR1,2 antibodies. Inset shows a greyscale image of the TbMORN1 channel with enhanced levels so that the endogenous protein is visible. Three separate clones were assayed, each in three independent experiments; an exemplary blot is shown. One of the three clones appeared to have slightly leaky expression, with TbMORN1 levels in the -Tet condition being higher than controls. **(B)** Quantification of overexpression. TbMORN1 levels in control, uninduced (-Tet) and induced (+Tet) were normalised relative to the loading control and expressed relative to the control cells. Approximately 7-fold overexpression was achieved

relative to control cells. Data were obtained from blots using 3 separate clones, each induced in 3 independent experiments. Bars show mean + SD. **(C)** Overexpression of untagged TbMORN1 is deleterious. Uninduced (-Tet) and TbMORN1 overexpressing (+Tet) cells were assayed at 24 h intervals in a 3-day time course. Data were obtained from blots using 3 separate clones, each induced in 3 independent experiments. Mean +/- SD.
(TIF)

**S8 Fig. Optimisation of digitonin fractionation. (A)** Schematic of the one-step fractionation scheme. Cells expressing cytosolic GFP (Batram et al., 2014) were incubated for 10 min with increasing concentrations (0–40 μg/ml) of digitonin, using 1% IGEPAL as a positive control for full extraction. At the end of the incubation, the soluble and insoluble fractions were separated by centrifugation. **(B)** Equal fractions (5%) of the supernatant (SN) and pellet (P) were probed with antibodies specific for GFP and the endoplasmic reticulum chaperone BiP. At 40 μg/ml digitonin, good solubilisation of GFP was achieved with only negligible solubilisation of BiP (arrow 1). Both proteins were efficiently solubilised using 1% IGEPAL as a positive control. Multiple independent experiments were carried out; an exemplary blot is shown. Note that the membrane was cut into strips prior to blotting, but the samples shown are from the same experiment. **(C)** Quantification of the immunoblot shown in B. **(D, E, F)** As per panels A-C, but with constant 40 μg/ml digitonin concentration and varying incubation time (0–30 min). Increasing the incubation time over a range of 15–30 min did not appear to notably increase the amount of GFP extraction. As a result, a 25 min incubation time was used in the subsequent experiments.
(TIF)

**S9 Fig. Comparison of MORN1 proteins, and secondary structure analysis of apicomplexan MORN1s. (A)** Amino acid sequence alignment of MORN1 proteins from *Trypanosoma brucei*, *Toxoplasma gondii* and *Plasmodium falciparum*. The number of amino acids in each protein is indicated, amino acid numbers in the alignment are those for TbMORN1. The alignment is coloured according to the amino acid properties. **(B)** Pairwise comparison of percentage sequence identity between the three proteins. **(C)** Far-UV CD measurements obtained for TgMORN1(1–15) and TgMORN1(7–15). The secondary structure content predictions for each of the measured proteins were calculated in BeStSel and are shown below the CD graph. **(D)** As (C), but PfMORN1(1–15), (2–15) and (7–15). Like TbMORN1, TgMORN1 and PfMORN1 are also all-β proteins.
(TIF)

**S10 Fig. Comparison of MORN1 extended dimers, and with SETD7 (SET7/9). (A)** TbMORN1(7–15) P2$_1$, TbMORN1(7–15) C2, and TgMORN1(7–15) extended dimers superimposed on each other and displayed in two orientations. In contrast to the other two proteins, the P2$_1$ crystal structure of TbMORN1(7–15) displays a bend of approximately 30˚ among the sub-units. **(B)** TbMORN1(7–15) MORN repeats superimposed on three MORN repeats from SETD7 (SET7/9). Alignment of the three MORN repeats from SETD7 with MORN repeat 7 from the TbMORN1(7–15) crystal structure over 22–23 aligned C-atoms yielded rmsd values of 2.3, 1.5 and 1.9 Å respectively. **(C)** Sequence alignment of MORN repeats from the TbMORN1(7–15) crystal structure with three MORN repeats from SETD7. The first Gly residue is conserved in all MORN repeats of TbMORN1(7–15) and SETD7 structures.
(TIF)

**S11 Fig. Conservation and properties of residues in TbMORN1(7–15). (A)** Conservation map of the TbMORN1(7–15) P2$_1$ crystal structure reveals a highly conserved stretch of residues along the groove. The structure is shown in two orientations, with residues colour-coded

according to the level of conservation. **(B)** An electrostatic map of TbMORN1(7–15) P2$_1$. Calculations were performed using APBS suite, displayed by Chimera. Colour scale: red = -13 kT; blue = +13 kT. Individual residues contributing to its surface electrostatics are labelled, namely those of the two negatively-charged loops building up a negative patch inside the groove, and the residues contributing to a small positively-charged region close to the dimer interface. **(C)** Hydrophobic map of TbMORN1(7–15) P2$_1$. Colour scale: blue = hydrophilic; orange = hydrophobic, pink = methionine residues.
(TIF)

**S12 Fig. MORN dimer interfaces and SAXS analysis of proteins in solution. (A)** Dimer interface of TbMORN1(7–15) C2 crystal form. In comparison to the P2$_1$ form, the dimer interface of C2 structure is broader, and is additionally stabilised by two disulphide bridges formed between Cys351 at the C-terminus of repeat 15 and Cys282 from the β-hairpin loop of repeat 12. **(B)** Crystal structure of the TgMORN1(7–15) V-shaped dimer, incorporating Zn$^{2+}$ in its dimerisation interface. **(C)** Dimer interface of the TgMORN1(7–15) extended dimer, which utilises residues from MORN repeats 13–15. In contrast to TbMORN1(7–15), where the dimerisation interface is centred around aromatic stacking, a hydrophobic core plays a crucial role in the dimer interface of extended TgMORN1(7–15). Leu327, Leu329, Leu335, Leu344, Val345, Val347, Phe350 and Phe352 are part of this hydrophobic core. The dimer is stabilised by a single salt bridge formed between the Asp308 of one subunit and the His333 of the other subunit. This salt bridge is further stabilised by two hydrogen bonds between the main-chain nitrogen of Val345 and a carbonyl oxygen of Val347 of respective subunits. **(D)** Dimer interface of the TgMORN1(7–15) V-shaped dimer. Cys305 and Asp308 incorporate a structural Zn$^{2+}$ ion, which stabilises the somewhat smaller dimerisation interface of this protein. Although its dimerisation interface is very similar to that of PfMORN1(7–15), it lacks the aromatic core of PfMORN1(7–15). The latter is replaced by a series of unique aromatic stacking interactions at the protein´s vertex, these being contributed by a pair of Phe350 residues, sandwiched between a pair of His333 residues. **(E)** Kratky plots derived from SAXS analysis of TbMORN1(7–15), TgMORN1(7–15), PfMORN1(7–15), and TbMORN1(2–15). The shape of the plots suggests an elongated shape of the dimers in solution.
(TIF)

**S13 Fig. Conservation of residues for formation of V-shaped dimers in Apicomplexa and related clades.** Amino acid sequence alignment of the C-termini of TbMORN1, TgMORN1, PfMORN1, and fifteen other MORN repeat-containing proteins from related taxa. Amino acid numbers are given according to the TbMORN1 protein, and the three proteins with experimentally-determined high-resolution structures are shown in bold within the black box. Essential for formation of a V-shaped dimer are a coordinating Cys residues and an anion-π interaction pair. In TbMORN1, the coordinating Cys residue has been substituted for Leu (Leu 301). Similarly, the Phe and Glu residues of the anion-π interaction pair (indicated with black arrows) have been substituted for Leu (Leu299) and are not present (deletion after Pro302) respectively. This supports the conclusion that TbMORN1 exists only in the extended form, while the apicomplexan proteins and those from related clades are probably capable of adopting both extended and V-shaped conformations.
(TIF)

**S1 Table.**
(TIF)

**S2 Table.**
(TIF)

**S3 Table.**
(TIF)

**S1 Video.**
(MP4)

**S1 Raw images.**
(PDF)

**S1 File.**
(PDF)

**S2 File.**
(PDF)

**S3 File.**
(PDF)

**S4 File.**
(PDF)

**S5 File.**
(PDF)

## Acknowledgments

We are indebted to Graham Warren (MRC Laboratory for Molecular Cell Biology, University College London), for comments and support. We are grateful to Sonja Lorenz (University of Würzburg) for critical reading of the manuscript. Tuguldur Tumurbaatar assisted with copy-editing and formatting. Valeria Stefania carried out the SAXS measurements. The Martens group at the Max Perutz Labs Vienna provided the Doc2B protein; Iva Lucic in the Leonard group supplied the Akt protein. Euripedes Almeida Ribeiro advised on fluorescence anisotropy measurements. Markus Engstler (University of Würzburg) provided the bloodstream form GFP cells as well as lab space and support. Susanne Fenz (University of Würzburg) assisted with the purification of trypanosome lipids. The authors thank the staff of MX and SAXS beamlines at ESRF in Grenoble and EMBL-Hamburg for their excellent support. EM data were recorded at the EM Facility of the Vienna BioCenter Core Facilities GmbH (VBCF), a member of the Vienna BioCenter (VBC), Austria. Proteomics analyses were performed by the Mass Spectrometry Facility at Max Perutz Labs Vienna using the VBCF instrument pool.

## Author Contributions

**Conceptualization:** Sara Sajko, Brooke Morriswood, Kristina Djinovic-Carugo.

**Data curation:** Brooke Morriswood, Kristina Djinovic-Carugo.

**Formal analysis:** Sara Sajko, Terry K. Smith, Brooke Morriswood, Kristina Djinovic-Carugo.

**Funding acquisition:** Anne-Claude Gavin, Thomas A. Leonard, Dimitri I. Svergun, Terry K. Smith, Brooke Morriswood, Kristina Djinovic-Carugo.

**Investigation:** Sara Sajko, Irina Grishkovskaya, Julius Kostan, Melissa Graewert, Kim Setiawan, Linda Trübestein, Korbinian Niedermüller, Charlotte Gehin, Antonio Sponga, Martin Puchinger, Anne-Claude Gavin, Thomas A. Leonard, Dimitri I. Svergun, Terry K. Smith, Brooke Morriswood, Kristina Djinovic-Carugo.

**Methodology:** Sara Sajko, Irina Grishkovskaya, Julius Kostan, Melissa Graewert, Kim Setia-wan, Linda Trübestein, Korbinian Niedermüller, Charlotte Gehin, Antonio Sponga, Martin Puchinger, Anne-Claude Gavin, Thomas A. Leonard, Dimitri I. Svergun, Terry K. Smith, Brooke Morriswood, Kristina Djinovic-Carugo.

**Project administration:** Brooke Morriswood, Kristina Djinovic-Carugo.

**Resources:** Thomas A. Leonard, Dimitri I. Svergun, Brooke Morriswood, Kristina Djinovic-Carugo.

**Supervision:** Anne-Claude Gavin, Thomas A. Leonard, Brooke Morriswood, Kristina Djino-vic-Carugo.

**Validation:** Sara Sajko, Julius Kostan, Brooke Morriswood, Kristina Djinovic-Carugo.

**Visualization:** Sara Sajko, Irina Grishkovskaya, Julius Kostan, Melissa Graewert, Terry K. Smith, Brooke Morriswood, Kristina Djinovic-Carugo.

**Writing – original draft:** Sara Sajko, Terry K. Smith, Brooke Morriswood, Kristina Djinovic-Carugo.

**Writing – review & editing:** Sara Sajko, Anne-Claude Gavin, Thomas A. Leonard, Terry K. Smith, Brooke Morriswood, Kristina Djinovic-Carugo.

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
