## [Decision Letter · Decision Letter 0]

1 Sep 2020

PONE-D-20-23015

Structures of three MORN repeat proteins and a re-evaluation of the proposed lipid-binding properties of MORN repeats

PLOS ONE

Dear Dr. Morriswood,

Thank you for submitting your manuscript to PLOS ONE. After careful consideration, we feel that it has merit but does not fully meet PLOS ONE’s publication criteria as it currently stands. Therefore, we invite you to submit a revised version of the manuscript that addresses the points raised during the review process.

All three reviewers agreed that the work presented in this manuscript is solid and sound, and the presentation of the data is clear. Overall, the quality of the work is high. They only asked a few editorial changes to improve the clarity of the manuscript. Please refer to the comments from individual reviewers for more details. I am sorry that it took a while to make an editorial decision. I appreciate the authors' patience during the reviewing process. 

We look forward to receiving your revised manuscript.

Kind regards,

Zhicheng Dou, Ph.D.

Academic Editor

PLOS ONE

Journal Requirements:

2.We note that you have indicated that data from this study are available upon request. PLOS only allows data to be available upon request if there are legal or ethical restrictions on sharing data publicly. For more information on unacceptable data access restrictions, please see http://journals.plos.org/plosone/s/data-availability#loc-unacceptable-data-access-restrictions.

3.PLOS ONE now requires that authors provide the original uncropped and unadjusted images underlying all blot or gel results reported in a submission’s figures or Supporting Information files. This policy and the journal’s other requirements for blot/gel reporting and figure preparation are described in detail at https://journals.plos.org/plosone/s/figures#loc-blot-and-gel-reporting-requirements and https://journals.plos.org/plosone/s/figures#loc-preparing-figures-from-image-files. When you submit your revised manuscript, please ensure that your figures adhere fully to these guidelines and provide the original underlying images for all blot or gel data reported in your submission. See the following link for instructions on providing the original image data: https://journals.plos.org/plosone/s/figures#loc-original-images-for-blots-and-gels.

Additional Editor Comments (if provided):

All three reviewers agreed that the work presented in this manuscript is solid and sound, and the presentation of the data is clear. Overall, the quality of the work is high. They only asked a few editorial changes to improve the clarity of the manuscript. Please refer to the comments from individual reviewers for more details. I am sorry that it took a while to make an editorial decision. I appreciate the authors' patience during the reviewing process.

Reviewers' comments:

Reviewer's Responses to Questions

**Comments to the Author**

1. Is the manuscript technically sound, and do the data support the conclusions?

Reviewer #1: Yes

Reviewer #2: Yes

Reviewer #3: Yes

2. Has the statistical analysis been performed appropriately and rigorously? 

Reviewer #1: Yes

Reviewer #2: Yes

Reviewer #3: Yes

3. Have the authors made all data underlying the findings in their manuscript fully available?

Reviewer #1: Yes

Reviewer #2: Yes

Reviewer #3: Yes

4. Is the manuscript presented in an intelligible fashion and written in standard English?

Reviewer #1: Yes

Reviewer #2: Yes

Reviewer #3: Yes

5. Review Comments to the Author

Reviewer #1: Review on the manuscript „Structures of three MORN repeat proteins and a re-evaluation of the proposed lipid-binding properties of MORN repeats.“

The authors have sufficiently addressed most of the points of critique highlighted by the reviewers. The work is solid and sound, the presentation of the data is clear (although still a bit lengthy). It is good material for PloS ONE. However, before recommending the manuscript for publication, the authors should still address one remaining point by a more extended discussion.

As explained by the authors, the first MORN repeat was removed from the protein for biochemical studies, because preparations of the full-length TbMORN1(1-15) was polydisperse. I understand the inherent logic of this decision. However, it is important to realize that a) the first MORN repeat and b) oligomerization of the protein (leading to polydispersity) may be crucial for membrane binding. Often, membrane binding proteins depends on avidity effects.

Therefore, the potential role of the first MORN repeat and the oligomerization of TbMORN1 should be discussed more clearly. Upon reading the manuscript, I kept asking myself if not the most distal portions of TbMORN1 (the first repeat) would be crucial to interact (tightly or transiently) with a target membrane in order to mediate lipid exchange between different membranes. This is also because a potential role as a lipid exchanger along the identified longitudinal groove running through the entire protein does not seem entirely unlikely. Potentially a sequence analysis of the first MORN repeat would help discussion a potential role for thefirst MOPRN repeat in membrane interaction?

It would be interesting (but not neccessary) to know if TbMORN(2-15) and TbMORN(1-15) have the same of different impact on E. coli PE-levels. Also, it would be interesting to know the impact of Tb(MORN(1-14) on the E. coli PE-levels. If these data are available – can the authors comment on them?

Reviewer #2: In this work, Sajko, Grishkovskaya, Kostan, and colleagues have generated and analyzed the structures of 3 proteins comprising repeats of the MORN domain, which previously had been thought to be primarily involved in membrane targeting via the recognition of phospholipids. Using an extensive range of biophysical methods, they show that the MIORN domain is not able to bind to membranes but does have the capacity to bind to individual phospholipids. In a series of experiments in T. brucei cells, they show that TbMORN1 does not appear to bind to membranes or require membranes to remain localized to a cytoskeleton-associated structure. They also show that overexpression of TbMORN1 is toxic to the cells. While much of this data ends up providing negative results, the scope of the experimental methods employed, along with excellent rigor and quantitation, provide a very strong argument that the assumed function of the MORN domain should be reconsidered. This is a very important contribution on its own.

The authors subsequently provide a detailed analysis of the 3 structures they have generated, including mutational analysis to identify key residues involved in dimerizing the MORN protein. The structures also strengthen the consensus for the MORN repeat based of their structural data, which is an important contribution. They also identify an alternate “V-shaped” comformation of the apicomplexan MORN homologs and make some tantalizing links to the different states of the basal complex, which constricts during cell division. Their structural data also suggests a potential mechanism for oligomerizing the parasite MORN proteins via an incomplete MORN repeat at the N-terminus, which could allow the protein to form some of the structures that are observed in cells.

Overall, the quality of the work is very high. I do not think any additional experiments are warranted but I do have a few editorial comments.

There are a few cases where figure call-outs in the text do not refer to the correct figure, especially in the supplemental section. For example, there is no mention of Figure S3 in the text, but it is present in the most recent version of the manuscript.

Line 225-227: There is no mention of what the result of the FA experiments are, which is confusing because they subsequently couldn’t be validated? Some initial statement of what the results are should be mentioned first.

Fig 3, C,D, G, H: Should have labels for the Ty1-tagged MORN and the WT.

Line 647-654: Why is the PfMORN SAXS analysis mentioned separately from the Tb and Tg proteins? Could it not be docked into the molecular envelope?

Reviewer #3: In this work, Sajko et al. characterized the structures of three MORN repeat proteins (TbMORN1, TgMORN1, PfMORN1) and lipid-binding properties of truncated TbMORN1 with different number of MORN repeats. The work is thorough, rigorous, and well presented. Both positive and negative results are carefully analyzed and informative. The highlight of the work is the discovery of a dimerization interface that appear to be conserved among evolutionarily divergent groups, revealed by structures of TbMORN1(7-15), TgMORN1(7-15), PfMORN1(7-15). Also of interest is the lipid binding activities of TbMORN1 in vitro. This reviewer only has a few minor comments and suggestions.

1. Figure S6 D&E: Error bars should be included in the bar graphs, which represent results from multiple independent replicates.

2. Figure S3: TbMORN1(10-15) appears to bind to PI(4,5)P2 specifically, whereas TbMORN1(2-15) appears to bind to all three forms of PIs tested. This suggests that the headgroup (not just the aliphatic chains) does play a role in binding to different parts of the MORN1 protein. Also currently Figure S3 is not cited in the main text.

3. Figures S2&3: From the rebuttal letter for the Reviewer Commons review, it seems that the PIP2 binding data were moved to the supplementary material per previous reviewers' suggestion. I actually think that Figures S2&3 should be combined into one figure and moved to the main text. The data quality is high and shows definitively that TbMORN1 binds to free PIP lipids, even though it does not bind to liposomes. It supports the intriguing hypothesis that MORN repeat proteins might function as lipid carriers.

4. p19 (line 797-799): Two other references should be added- Heaslip et al. PLoS Pathogens 2010, 6(2): e1000754; Lorestani et al. PLoS One. 2010, 5(8):e12302.

6. PLOS authors have the option to publish the peer review history of their article (what does this mean?). If published, this will include your full peer review and any attached files.

Reviewer #1: No

Reviewer #2: No

Reviewer #3: No

---

## [Author Response · Author response to Decision Letter 0]

6 Nov 2020

Dear PLoS ONE,

Our responses to the points raised by the academic editor and reviewers are below in red text.

 Yours faithfully,

 Kristina Djinovic-Carugo, Brooke Morriswood, Sara Sajko

PONE-D-20-23015

Structures of three MORN repeat proteins and a re-evaluation of the proposed lipid-binding properties of MORN repeats

PLOS ONE

Dear Dr. Morriswood,

Thank you for submitting your manuscript to PLOS ONE. After careful consideration, we feel that it has merit but does not fully meet PLOS ONE’s publication criteria as it currently stands. Therefore, we invite you to submit a revised version of the manuscript that addresses the points raised during the review process.

All three reviewers agreed that the work presented in this manuscript is solid and sound, and the presentation of the data is clear. Overall, the quality of the work is high. They only asked a few editorial changes to improve the clarity of the manuscript. Please refer to the comments from individual reviewers for more details. I am sorry that it took a while to make an editorial decision. I appreciate the authors' patience during the reviewing process.

We are very glad that the reviewers were happy with the data and manuscript, and we have implemented virtually all of the suggested changes.

We look forward to receiving your revised manuscript.

Kind regards,

Zhicheng Dou, Ph.D.

Academic Editor

PLOS ONE

Additional Editor Comments (if provided):

All three reviewers agreed that the work presented in this manuscript is solid and sound, and the presentation of the data is clear. Overall, the quality of the work is high. They only asked a few editorial changes to improve the clarity of the manuscript. Please refer to the comments from individual reviewers for more details. I am sorry that it took a while to make an editorial decision. I appreciate the authors' patience during the reviewing process.

Reviewers' comments:

Reviewer's Responses to Questions

Comments to the Author

1. Is the manuscript technically sound, and do the data support the conclusions?

Reviewer #1: Yes

Reviewer #2: Yes

Reviewer #3: Yes

2. Has the statistical analysis been performed appropriately and rigorously? 

Reviewer #1: Yes

Reviewer #2: Yes

Reviewer #3: Yes

3. Have the authors made all data underlying the findings in their manuscript fully available?

Reviewer #1: Yes

Reviewer #2: Yes

Reviewer #3: Yes

4. Is the manuscript presented in an intelligible fashion and written in standard English?

Reviewer #1: Yes

Reviewer #2: Yes

Reviewer #3: Yes

5. Review Comments to the Author

Reviewer #1: Review on the manuscript „Structures of three MORN repeat proteins and a re-evaluation of the proposed lipid-binding properties of MORN repeats.“

The authors have sufficiently addressed most of the points of critique highlighted by the reviewers. The work is solid and sound, the presentation of the data is clear (although still a bit lengthy). It is good material for PloS ONE. However, before recommending the manuscript for publication, the authors should still address one remaining point by a more extended discussion.

As explained by the authors, the first MORN repeat was removed from the protein for biochemical studies, because preparations of the full-length TbMORN1(1-15) was polydisperse. I understand the inherent logic of this decision. However, it is important to realize that a) the first MORN repeat and b) oligomerization of the protein (leading to polydispersity) may be crucial for membrane binding. Often, membrane binding proteins depends on avidity effects.

Therefore, the potential role of the first MORN repeat and the oligomerization of TbMORN1 should be discussed more clearly. Upon reading the manuscript, I kept asking myself if not the most distal portions of TbMORN1 (the first repeat) would be crucial to interact (tightly or transiently) with a target membrane in order to mediate lipid exchange between different membranes. This is also because a potential role as a lipid exchanger along the identified longitudinal groove running through the entire protein does not seem entirely unlikely. Potentially a sequence analysis of the first MORN repeat would help discussion a potential role for thefirst MOPRN repeat in membrane interaction? 

The first MORN repeat is clearly involved in protein oligomerisation (see Fig. 1B), but the reviewer is absolutely right that this may also imply that the oligomer is the relevant species for membrane localisation/interaction. It is important to remember, however, that the in vivo data (Figs 3, 4) do not support membrane association of the overexpressed full-length protein, which partitions into the cytosolic fraction.

We have now explicitly noted in the Discussion that it remains an open possibility that the oligomer could indirectly (or transiently) associate with the plasma membrane, and once in proximity it could potentially be involved as a lipid exchanger (lines 831-833). 

This is an interesting hypothesis and one that we would like to pursue, but it is outside the frame of the present study and will probably have to wait until we have established conditions for the investigation of the oligomer in vitro. 

It would be interesting (but not neccessary) to know if TbMORN(2-15) and TbMORN(1-15) have the same of different impact on E. coli PE-levels. Also, it would be interesting to know the impact of Tb(MORN(1-14) on the E. coli PE-levels. If these data are available – can the authors comment on them? 

Unfortunately, we do not at present have any data on the impact of TbMORN1(2-15) or TbMORN1(1-14) on bacterial lipid profiles, so we are not able to comment. We agree that it would be interesting to look into this.

Reviewer #2: In this work, Sajko, Grishkovskaya, Kostan, and colleagues have generated and analyzed the structures of 3 proteins comprising repeats of the MORN domain, which previously had been thought to be primarily involved in membrane targeting via the recognition of phospholipids. Using an extensive range of biophysical methods, they show that the MIORN domain is not able to bind to membranes but does have the capacity to bind to individual phospholipids. In a series of experiments in T. brucei cells, they show that TbMORN1 does not appear to bind to membranes or require membranes to remain localized to a cytoskeleton-associated structure. They also show that overexpression of TbMORN1 is toxic to the cells. While much of this data ends up providing negative results, the scope of the experimental methods employed, along with excellent rigor and quantitation, provide a very strong argument that the assumed function of the MORN domain should be reconsidered. This is a very important contribution on its own.

The authors subsequently provide a detailed analysis of the 3 structures they have generated, including mutational analysis to identify key residues involved in dimerizing the MORN protein. The structures also strengthen the consensus for the MORN repeat based of their structural data, which is an important contribution. They also identify an alternate “V-shaped” comformation of the apicomplexan MORN homologs and make some tantalizing links to the different states of the basal complex, which constricts during cell division. Their structural data also suggests a potential mechanism for oligomerizing the parasite MORN proteins via an incomplete MORN repeat at the N-terminus, which could allow the protein to form some of the structures that are observed in cells.

Overall, the quality of the work is very high. I do not think any additional experiments are warranted but I do have a few editorial comments.

There are a few cases where figure call-outs in the text do not refer to the correct figure, especially in the supplemental section. For example, there is no mention of Figure S3 in the text, but it is present in the most recent version of the manuscript.

Thanks for spotting this, which occurred during the revision process. We have implemented the suggestion and checked all figure citations. There is now an extra sentence containing a citation of Figure S3.

Line 225-227: There is no mention of what the result of the FA experiments are, which is confusing because they subsequently couldn’t be validated? Some initial statement of what the results are should be mentioned first. 

We have implemented this suggestion. There is now an extra sentence to clarify the FA results. Binding was only seen for long-chain lipids.

Fig 3, C,D, G, H: Should have labels for the Ty1-tagged MORN and the WT.

Implemented. 

Line 647-654: Why is the PfMORN SAXS analysis mentioned separately from the Tb and Tg proteins? Could it not be docked into the molecular envelope? 

The PgMORN1 SAXS analysis was mentioned separately to that of the Tb and Tg proteins because data analysis showed that PfMORN1 adopts an extended structure in solution (Fig. 7C). An extended structure was docked into the molecular envelope, as shown in Fig 7C. The text has been amended to clarify this point. 

Reviewer #3: In this work, Sajko et al. characterized the structures of three MORN repeat proteins (TbMORN1, TgMORN1, PfMORN1) and lipid-binding properties of truncated TbMORN1 with different number of MORN repeats. The work is thorough, rigorous, and well presented. Both positive and negative results are carefully analyzed and informative. The highlight of the work is the discovery of a dimerization interface that appear to be conserved among evolutionarily divergent groups, revealed by structures of TbMORN1(7-15), TgMORN1(7-15), PfMORN1(7-15). Also of interest is the lipid binding activities of TbMORN1 in vitro. This reviewer only has a few minor comments and suggestions.

1. Figure S6 D&E: Error bars should be included in the bar graphs, which represent results from multiple independent replicates. 

Implemented.

2. Figure S3: TbMORN1(10-15) appears to bind to PI(4,5)P2 specifically, whereas TbMORN1(2-15) appears to bind to all three forms of PIs tested. This suggests that the headgroup (not just the aliphatic chains) does play a role in binding to different parts of the MORN1 protein. Also currently Figure S3 is not cited in the main text. 

We agree that TbMORN1(10-15) appears to bind to PI(4,5)P2, as does TbMORN1(2-15). We do not however agree that TbMORN1(2-15) is binding to PI and PI(4)P. 

No band shift was seen at all for PI(4)P (see attached annotation of Figure S3), unlike the clear shift seen for PI(4,5)P2. The case of PI is, we concede, more ambiguous but this is due to the gel having been run for a shorter time than that shown in panel B. 

We have now added a citation for Figure S3 (apologies - this was lost during the revision process).

3. Figures S2&3: From the rebuttal letter for the Reviewer Commons review, it seems that the PIP2 binding data were moved to the supplementary material per previous reviewers' suggestion. I actually think that Figures S2&3 should be combined into one figure and moved to the main text. The data quality is high and shows definitively that TbMORN1 binds to free PIP lipids, even though it does not bind to liposomes. It supports the intriguing hypothesis that MORN repeat proteins might function as lipid carriers. 

We would prefer not to implement this suggestion. Although we are very pleased that the reviewer is so positive about the data, we would prefer to leave things as they are. In the expanded description of Figures S2 and S3 we now mention that the data show definitively that TbMORN1 binds to free lipids.

4. p19 (line 797-799): Two other references should be added- Heaslip et al. PLoS Pathogens 2010, 6(2): e1000754; Lorestani et al. PLoS One. 2010, 5(8):e12302. Implemented.

---

## [Editor Report · Decision Letter 1]

9 Nov 2020

Structures of three MORN repeat proteins and a re-evaluation of the proposed lipid-binding properties of MORN repeats

PONE-D-20-23015R1

Dear Dr. Morriswood,

We’re pleased to inform you that your manuscript has been judged scientifically suitable for publication and will be formally accepted for publication once it meets all outstanding technical requirements.

Kind regards,

Zhicheng Dou, Ph.D.

Academic Editor

PLOS ONE
---

## [Editor Report · Acceptance letter]

16 Nov 2020

PONE-D-20-23015R1 

Structures of three MORN repeat proteins and a re-evaluation of the proposed lipid-binding properties of MORN repeats 

Dear Dr. Morriswood:

I'm pleased to inform you that your manuscript has been deemed suitable for publication in PLOS ONE. Congratulations! Your manuscript is now with our production department. 

Kind regards, 

on behalf of

Dr. Zhicheng Dou 

Academic Editor

PLOS ONE